# Detection of Fungi and Oomycetes by Volatiles Using E-Nose and SPME-GC/MS Platforms

**DOI:** 10.3390/molecules25235749

**Published:** 2020-12-05

**Authors:** Jérémie Loulier, François Lefort, Marcin Stocki, Monika Asztemborska, Rafał Szmigielski, Krzysztof Siwek, Tomasz Grzywacz, Tom Hsiang, Sławomir Ślusarski, Tomasz Oszako, Marcin Klisz, Rafał Tarakowski, Justyna Anna Nowakowska

**Affiliations:** 1InTNE (Plants & Pathogens Group), Hepia, University of Applied Sciences and Arts of Western Switzerland, 150 route de Presinge, 1254 Jussy, Switzerland; jeremie.loulier@bluewin.ch; 2Institute of Forest Sciences, Faculty of Civil Engineering and Environmental Sciences, Bialystok University of Technology, Wiejska 45E, 15-351 Bialystok, Poland; m.stocki@pb.edu.pl (M.S.); T.Oszako@ibles.waw.pl (T.O.); 3Institute of Physical Chemistry, Polish Academy of Sciences, Kasprzaka 44/52, 01-224 Warsaw, Poland; monika@ichf.edu.pl (M.A.); ralf@ichf.edu.pl (R.S.); 4Faculty of Electrical Engineering, Warsaw University of Technology, Koszykowa 75, 00-661 Warsaw, Poland; krzysztof.siwek@ee.pw.edu.pl (K.S.); tomasz.grzywacz@ee.pw.edu.pl (T.G.); 5Environmental Sciences, University of Guelph, Guelph, ON N1G 2W1, Canada; thsiang@uoguelph.ca; 6Forest Protection Department, Forest Research Institute, Braci Leśnej 3, 05-090 Sękocin Stary, Poland; S.Slusarski@ibles.waw.pl; 7Department of Silviculture and Genetics, Forest Research Institute, Braci Leśnej 3, 05-090 Sękocin Stary, Poland; M.Klisz@ibles.waw.pl; 8Faculty of Physics, Warsaw University of Technology, Koszykowa 75, 00-662 Warsaw, Poland; Rafal.Tarakowski@pw.edu.pl; 9Institute of Biological Sciences, Cardinal Stefan Wyszynski University in Warsaw, Wóycickiego 1/3 Street, 01-938 Warsaw, Poland

**Keywords:** fungi, oomycetes, VOCs, EI mass spectrometry, SPME-GC/MS, e-nose, sesquiterpenes

## Abstract

Fungi and oomycetes release volatiles into their environment which could be used for olfactory detection and identification of these organisms by electronic-nose (e-nose). The aim of this study was to survey volatile compound emission using an e-nose device and to identify released molecules through solid phase microextraction–gas chromatography/mass spectrometry (SPME–GC/MS) analysis to ultimately develop a detection system for fungi and fungi-like organisms. To this end, cultures of eight fungi (*Armillaria gallica*, *Armillaria ostoyae*, *Fusarium avenaceum*, *Fusarium culmorum*, *Fusarium oxysporum*, *Fusarium poae*, *Rhizoctonia solani*, *Trichoderma asperellum*) and four oomycetes (*Phytophthora cactorum*, *P. cinnamomi*, *P. plurivora*, *P. ramorum*) were tested with the e-nose system and investigated by means of SPME-GC/MS. Strains of *F. poae*, *R. solani* and *T. asperellum* appeared to be the most odoriferous. All investigated fungal species (except *R. solani*) produced sesquiterpenes in variable amounts, in contrast to the tested oomycetes strains. Other molecules such as aliphatic hydrocarbons, alcohols, aldehydes, esters and benzene derivatives were found in all samples. The results suggested that the major differences between respective VOC emission ranges of the tested species lie in sesquiterpene production, with fungi emitting some while oomycetes released none or smaller amounts of such molecules. Our e-nose system could discriminate between the odors emitted by *P. ramorum*, *F. poae*, *T. asperellum* and *R. solani*, which accounted for over 88% of the PCA variance. These preliminary results of fungal and oomycete detection make the e-nose device suitable for further sensor design as a potential tool for forest managers, other plant managers, as well as regulatory agencies such as quarantine services.

## 1. Introduction

Many phytopathogenic fungi and oomycetes species causing damping off and root rot diseases, are known to emit several secondary metabolites in the form of volatile organic compounds (VOCs), which can be genus- or species-specific. Other microbial secondary metabolites include, for instance, antibiotics, enzymes and toxins. Even though a great diversity of VOCs can be found from many different species, some volatiles are unique to particular species [1,2]. Volatile organic compounds are carbon molecules of molecular mass classically ranging from 100 to 500 Da and chain size up to C_20_ that occur in gaseous form above 20 °C and 10 Pa [3,4]. Generally lipophilic and thus poorly soluble in water, their boiling point is low and their vapor pressure at room temperature is high. Many of them have an associated characteristic odor [5].

Fungi produce VOC mixtures through primary and secondary metabolic pathways, and these diffuse through soil and into the atmosphere [6]. VOC production has been investigated in only ca. 100 of the over 100,000 described fungal species to date [7]. Fungal VOCs released through secondary metabolism seem to be at their highest during sporulation and mycotoxin production [8]. The secreted mixture profile varies according to many factors linked with the fungal strain itself and its growth environment, including physiological state, temperature, pH, oxygen, nutrients, incubation time, etc. [9,10].

Around one thousand different volatile compounds have to date been identified in a broad spectrum of microorganisms [11], produced mainly through glucose oxidation from diverse intermediates [6]. These molecules participate in various pathways of primary and secondary metabolism such as aerobic heterotrophic carbon metabolism, amino-acid catabolism, fatty acids degradation, fermentation, sulphur reduction and terpenoids biosynthesis [12]. Among these VOCs, more than one quarter are of unique fungal origin and include alcohols, aldehydes, alkenes, benzenoids, ketones, esters, terpenes and sulfuric compounds, although many other families are found as well, e.g., simple hydrocarbons, furans, heterocycles, phenols, thioalcohols, thioesters, etc. [6,13,14].

Sesquiterpenes are terpene hydrocarbons with an identical molecular formula of C_15_H_24_. Their skeletons are made of three isoprene units and can be acyclic or may include rings. Generally, sesquiterpenoids derive from sesquiterpenes through molecular rearrangements or oxidation. Sesquiterpene emission by fungi has been a field of particular interest over the past years [11,15], because this group of compounds may be key in discriminating between soil-borne fungi and *Phytophthora* spp. [16].

The exact reasons why microorganisms emit volatiles are not fully understood. It has been proposed that these molecules could be waste products resulting from microbial metabolism, and released for detoxification purposes [17]. Previous studies have shown that volatiles are involved in antimicrobial activity [8,18]. They have highlighted VOC involvement in interactions with distant microbial colonies in the form of infochemical compounds, influencing development, gene expression, and behavior of the recipient microorganisms [4]. During the last fifteen years, many trials were implemented in different research fields (for instance in environmental, food or medicinal sciences) to increase our understanding of microbial volatiles, facilitated by the development of efficient detection methods [19].

The first attempts of vapor characterization using sensing devices that can be considered as e-noses date back to 1960 [20], and they showed promise as tools for characterization of complex fungal VOC mixtures [21]. In general, an e-nose can refer to a wide range of devices including a set of gas sensors coupled with an information processing system and recognition software, founded on theoretical models as well as reference literature [22]. It was only in 1982 that intelligent e-noses able to classify odors really began to be developed [23], with the “electronic nose” appellation emerging for the first time [24]. A recent study managed to detect low concentrated matsutake alcohol by means of an e-nose using surface acoustic wave technology, which makes it a promising candidate for future developments in fungal e-nose detection [25].

In phytopathology, pathogen identification is usually achieved through study of symptoms, pathogen isolation, morphological description and possibly immunological testing or DNA sequence analyses. However, infection symptoms are sometimes invisible, hidden underground or non-specific for the pathogenic agent, which complicates proper diagnosis. Pathogen isolation, immunological tests, and DNA tests require laboratory work making the operation expensive, laborious and time-consuming, thus not well suited for large-scale monitoring [26]. In this regard, e-nose could serve as a preliminary screening tool allowing a quick diagnosis that could be achieved even without any symptoms being visible on the plant, as well as an efficient field monitoring applicable at the scale of single seeds, entire plants, or food storage installations.

To improve the performance of this technology for application to forest sciences or related fields, identification of pathogen-specific VOC profiles could be a key point for improved e-nose sensing efficiency. The aim of this work was to test an electronic nose developed by the Warsaw University of Technology, originally designed to identify bacterial and fungal rot in food, to the field of phytopathology. We focused on detection of eight well-known fungal pathogens i.e., *Armillaria* spp., *Fusarium* spp., and *Rhizoctonia solani* causing root rot damage e.g., in Scots pine, black alder and European oak seedlings [27]. The other tested fungus (*Trichoderma asperellum*) is not pathogenic and rather considered as a biocontrol agent in many forest and ornamental nurseries [27]. Furthermore, four *Phytophthora* species were investigated, since these are recognized as soil-borne pathogens of many forest tree species, including European oak, European beech, silver birch and black alder [28]. Two working hypotheses were tested. First, we assumed that the e-nose apparatus can be a suitable detection system for fungi and fungi-like organisms. The second hypothesis was that some VOCs are species-specific and because these VOCs emitted by microorganisms can be detected, this can be used to differentiate the investigated microbial species. To test these hypotheses we did the following: (i) selected and developed pure cultures of the studied microorganisms; (ii) investigated the gas composition in the air with e-nose technology; (iii) performed a precise detection of VOCs using two independent SPME-GC/MS analytical methods, and finally (iv) speculated on the extent the emitted compounds are useful for identification of a given microorganism.

## 2. Results

### 2.1. E-Nose Gas Detection

Distribution of the overall averaged values of gas detected by twelve differential sensor signals differed among all fourteen treatments (Figure 1). The two control treatments yielded somewhat divergent results because mean signal intensities obtained over four weeks of study showed clean empty glass flasks to be inducing a very weak response from e-nose sensors, whereas the potato dextrose agar (PDA) medium generated a stronger reaction from the sensing device on average (Figure 1a,b).

The e-nose detection showed that *A. gallica* and *A. ostoyae* cultures emitted low levels of volatiles similar to the empty flask control (Figure 1c,d). All *Fusarium* species provoked a clear reaction from e-nose sensors, with a slightly higher signal strength variation among replicates (Figure 1e–h), but still at the level comparable to the signal intensity for PDA medium (Figure 1e–g). The overall odor was thus moderate, except for *F. poae* which appeared to be one of the most odoriferous tested strains (Figure 1h), with an immediate and powerful rise in sensor signal.

*Phytophthora* species generated an overall feeble reaction from the e-nose (Figure 1i–l), except for *P. ramorum* which produced a strong odor (Figure 1l), only slightly weaker than *F. poae*, *R. solani* and *T. asperellum*. Signal strength usually increased quickly after the e-nose air inlet was introduced into the *P*. *ramorum* atmosphere. Like *F. poae*, *R. solani* and *T. asperellum* emitted a very strong odor highlighted during e-nose testing, with the introduction of the air inlet into a sample atmosphere which made the sensor signals instantly and strongly increase (Figure 1m,n). To sum up, the strains of *F. avenaceum*, *F. culmorum* and *F. oxysporum* along with *P. ramorum* generated less intense average reactions from the sensing device (Figure 1e–g,l), but these were however still stronger than for *Armillaria* and remaining *Phytophthora* strains whose odors appeared to be very weak (Figure 1c,d,i–k).

No clear pattern over time could be found among the tested species, probably because of the low level of repeated sampling (usually 3) per week of study which did not allow adequate assessment of temporal variation in emissions. The PDA medium was more constant between repeated observations, although the signal strength increased over time, possibly as the medium dried out. Sensors 1, 2, 3, 4, 11 and 12 overall systematically gave a stronger response to volatiles, while sensors 5, 6, 7 and even more 8, 9, 10 yielded weaker signals (Figure 1).

Based on correlation analyses between eigenvalues, we found that the variance between signals mostly depended on the first principal component (PC1) and only slightly on the second principal component (PC2) (93.87% and 03.26%, respectively; Figure 2, Appendix A). This enabled us to deduce using the PCA plot that the odors generated by *P. ramorum*, *F. poae*, *T. asperellum* and *R. solani* were readily distinguishable due to the location of the microorganisms on the biplot and grouping pattern confirmed by hierarchical clustering analysis (Appendix A, Appendix A).

### 2.2. Determination of VOC Emitted by Fungi and Oomycetes

In our study, two SPME fibers were compared (PDMS vs. PDMS/Carboxen). The results revealed that the PDMS fiber system was less applicable for volatiles released by fungi and oomycetes, because during GC/MS analyses a number of silicone-derived contaminants were observed. The latter could be explained by the chemical destruction of a DMS surface coating by biogenic volatiles released. Hence, only results obtained from the sampling of volatiles with a DMS/Carboxen fiber are presented.

The molecular identification was carried out using threefold approach: matching with GC/EI MS libraries, both commercial (NIST + Willey) and in-house built, comparison of Kovats indices (RI) (measured + reported elsewhere) and authentic standards. The last approach was applied only when standards were available, and thus was confined to common organics, such as hexanal, heptanal or benzaldehyde. However, a few unknowns from fungal emissions, e.g., 1,3,4,5,6,7-hexahydro-2,5,5-trimethyl-2*H*-2,4a-ethanonaphthalene, were tentatively elucidated using the two first approaches, because of the lack of authentic standards. The identification with standards (authentic not internal) is the gold standard for the thorough identification.

To identify components, both mass spectral data and the calculated retention indices were used. Mass spectrometric identification not confirmed by the retention index was considered as tentative. The background emissions from controls, empty containers and PDA medium were subtracted. Some of the volatiles found through GC/MS, such as aliphatic hydrocarbons, alcohols, aldehydes, esters and aromatic derivatives were present in samples from all treatments (including controls), but all the investigated fungi species (except *R. solani*) produced sesquiterpenes of variable quantities. However, the tested oomycete strains generally produced fewer detectable substances, even less than non-inoculated controls, including those compounds mentioned above (Table 1).

#### 2.2.1. Control Treatments

The empty container control atmosphere appeared to be surprisingly rich in ten volatiles (Table 1, treatment 1), whose origin was probably linked to the silicone ring adjoined to the underside of the lid by the manufacturer to allow hermetic closure of the flask. This was consistent with the e-nose sensing a very weak, but still detectable, signal for this control treatment (Figure 1a). Even though the e-nose sensors showed more signals from PDA medium control samples than the empty container control ones (Figure 1b), respective molecule abundances of both chromatograms belonged to the same order of magnitude (10^7^).

#### 2.2.2. *Armillaria* Species

The SPME-GC/MS data indicated a similar emission pattern in two *Armillaria* treatments which is in accord with the results obtained from e-nose analysis (Figure 1c,d). Two *Armillaria* species shared three similar VOCs i.e., (2*E*)-4,4-dimethyl-2-pentenal, (*E*)-2-octen-1-al, and 1,3,4,5,6,7-hexahydro-2,5,5-trimethyl-2*H*-2,4a-ethanonaphthalene (Table 1, treatments 3 and 4). Some sesquiterpenes, i.e., δ-cadinene, (*Z*)-α-bisabolene, and 1,3,4,5,6,7-hexahydro-2,5,5,-trimethyl-2H-2,4a-ethanonaphthalene were present only in *Armillaria* atmosphere suggesting an inherent emission by the fungi (Table 1, treatments 3 and 4). Another sesquiterpene, β-barbatene, was detected in both *A. ostoyae* (a basidiomycete) and *F. poae* (an ascomycete) implying common VOC emission (Table 1, treatment 4 and 8) between two different fungal divisions.

#### 2.2.3. *Fusarium* Species

The three *Fusarium* species showed an interesting pattern of VOC emission. Longifolene was detected among the most common volatiles in only *F. avenaceum* and *F. poae* (Table 1, treatments 5 and 8). Subsequently, production of β-phellandrene by *F. avenaceum* and *F. culmorum* was observed. α-pinene and Δ-3-Carene were detected in *P. plurivora* (Table 1, treatment 11). These compounds were not found in the non-inoculated media controls. Finally, *F. culmorum* contained ethanol, in common with *P. cinnamomi* and *P. ramorum* (Table 1, treatments 6, 10 and 12). Consistent with e-nose analysis results, the *F. poae* GC/MS data showed a higher molecular abundance magnitude (10^8^) (and remarkable compound diversity) than other *Fusarium* species (usually at 10^7^). The compound 1-octen-3-ol was found in *F. poae*, *P. cactorum, P. cinnamomi* and *R. solani* (Table 1, treatments 8, 9 and 13), supporting the idea of common molecules emitted across varied taxa.

#### 2.2.4. *Phytophthora* Species

Acetone was shared between two oomycetes *P. cactorum* and *P. plurivora* while ethanol was shared between *P. cinnamomi* and *P. ramorum* (Table 1, treatments 9 and 11). Other volatiles were present in different species of oomycetes and fungi, e.g., 3-octanone detected in *P. cactorum, P. ramorum* and *R. solani* (Table 1, treatment 9 and 13), and 1-octen-3-ol found in the *P. cactorum*, P. ramorum, *F. poae* and *R. solani* (Table 1, treatments 8, 9 and 13). The α-pinene and Δ-3-Carene emitted by *P. plurivora* were the only terpene compounds found in a *Phytophthora* species during this research (Table 1, treatment 11). Abundance of *P. ramorum* compounds was higher than for other *Phytophthora* sp., in accordance with e-nose analysis results. Among the studied Oomycetes, only *P. ramorum* showed the highest level and similar order of overall VOC emission as *Armillaria* treatments (Table 1, treatments 3, 4 and 13). SPME-GC/MS analysis revealed presence of 2-phenylethanol in both *P. ramorum* and *T. asperellum* (Table 1, treatments 12 and 14).

#### 2.2.5. *Rhizoctonia* and *Trichoderma* Species

The *R. solani* chromatogram differed from the signal indicated by e-nose sensors (Figure 1m). Higher amounts of 1-octen-3-ol were found in *R. solani* than in *F. poae* and *P. cactorum* (Table 1, treatments 8, 9 and 13). Consistent with e-nose results (Figure 1n), SPME-GC/MS analysis yielded for *T. asperellum* high richness of VOCs emitted with relatively high abundance magnitude (almost 10^8^).

## 3. Discussion

The sustainability (durability) and biodiversity of forest stands in Poland depends not only sustainable forest management but also on integrated pest management. Foresters cannot afford to allow pests to escape from nurseries to forest plantations together with the sold seedlings. Visual inspection often fails (asymptomatic plants), so forest managers are looking for new tools to support their work. In particular, pesticides applied in nurseries mask diseases, which in suitable conditions after outplanting will start to develop e.g., *Phytophthora* in wet forest sites. New devices should allow a quick detection of potential pathogens, particularly for emerging diseases. Among them are the foreign, invasive oomycetes (*Phytophthora*, *Pythium*) and other soil borne pathogens in the genera *Fusarium*, *Rhizoctonia* or *Cylindrocarpon*. All those organisms growing in pure cultures have distinctive strong odors because of volatile secondary metabolite production, but no instrument has been used in practice for routine detection of pathogen by forest managers. In the present research, we focused on assessing the possibility of designing an electronic nose to recognize genera or species of some pathogenic organisms often found in forest nurseries. This e-nose apparatus was made to recognize the specific electronic footprints produced by a VOC mixture interacting with a set of sensors. The change in physicochemical properties of the sensors induced by interaction with VOCs is transduced in a characteristic electrical signal, which allows description of the compounds without having to isolate the different components of the mixture [43,44]. The instrument used in this study has been previously used for recognition of very diverse volatile emission sources such as gasoline, coffee, tobacco and even explosives [45,46,47,48,49]. Certainly, our laboratory studies on volatile emission by microorganisms were carried out under optimal growth conditions which may differ from natural environments [50]. Moreover, the volatile mixture produced by a mixed colony of microorganisms is very likely to differ from what a pure culture may release [2].

A comparison of substances found in our samples (and an examination of where else it has been found previously) allowed us to screen substances (that are listed in Table 1), and to point out those which were specific to particular organisms. A major goal was to elucidate compounds that were revealed, and afterwards describe where they have been previously found and what significance they might have.

### 3.1. Accurateness of Analyses Performed by E-Nose

PCA revealed a clear distinction between the four fungal species detected by e-nose measurements, i.e., *P. ramorum*, *F. poae*, *T. asperellum* and *R. solani* compared to the controls. In strawberry for instance, a strong separation of pathogenic fungi i.e., *Botrytis* sp., *Penicillium* sp. and *Rhizopus* sp. was also based on the first two components of a PCA plot (accounting for 99.4% of variance) [51]. PCA of sensorial measurements under laboratory conditions also highlighted a strict relationship between the disease severity (potato brown rot) and the responses of the e-nose sensors [52].

A SPME-GC/MS revealed that sensors 1 and 2 were receptive to ethanol (for which GC/MS analysis detected it in *F. culmorum*, *P. cinnamomi* and *P. ramorum* with high confidence). Sensors 3, 4, 11 and 12 were receptive to VOCs in general (sesquiterpenes were frequently observed as well as other VOCs during GC/MS), and this may explain the high average corresponding signals observed. In contrast, the other sensors 5, 6, 9 and 10 may have been reacting to aliphatic hydrocarbons, as their sensitivity range included liquefied petroleum gases such as propane, butane, etc. On the other hand, sensors number 7, 8, 9 and 10 were sensitive to methane (7 and 8) or to methane and LP gases (9 and 10) and provided overall weak signals, which may have led to the conclusion that methane did not belong to the main compounds emitted by the tested samples. Furthermore, methane was not found in any sample during SPME-GC/MS.

### 3.2. Volatiles Identifying Fungal and Oomycete Species

#### 3.2.1. Control Treatments

##### Empty Flask Control

SPME-GC/MS analysis yielded a low abundance for the empty flask control chromatogram (order of magnitude 10^7^), compared to some other treatments. However, the empty flask control atmosphere contained some volatiles, which was consistent with the e-nose sensing a very weak, but still detectable, signal for this control treatment (Figure 1a). 2-ethyl-1-hexanol, tetradecane, and butylated hydroxytoluene are all compounds that may be found in adhesives, glues, gums, mastics, waxes, etc. For instance, 2-ethyl-1-hexanol is among the most highly produced worldwide synthetic alcohols and often serves as a solvent, while its ester derivatives have many industrial usages: adhesives, coatings, defoamers, emollients, inks, lubricants, plasticizers, etc. [53,54]. Paraffin wax and similar substances are frequently used as lubricants, insulators, water repellents or coating agents by industry. These include long-chain aliphatic alkanes whose oxidation by sample air could explain the presence of found aliphatic aldehydes (heptanal, octanal, nonanal, decanal and dodecanal). All these compounds were probably emitted by the silicone ring adjoined to the inner side of the lid by the manufacturer to allow hermetic closure of the flask, which thus appeared not to be chemically inert. Exposure to high temperatures during sterilization may have caused or facilitated volatile emission.

##### PDA Medium Control

The order of magnitude of the PDA control chromatogram was equivalent to what was obtained for the empty flask control (10^7^), even though the e-nose analysis showed the former to release higher levels of detectable odors than the latter. Nonanal, decanal, dodecanal and butylated hydroxytoluene were found here again, suggesting these compounds may be released by the flask and not the nutritive medium itself. Furthermore, several of the molecules detected in PDA control that were not detected from the empty flask (benzaldehyde, dodecane, etc.) may also have arisen from the glue, rubber and other synthetic materials [55], making it uncertain the exact origin of these observed volatiles.

#### 3.2.2. *Armillaria* Species

In accordance with e-nose analysis results, SPME-GC/MS indicated a similar chromatogram signal intensity between controls and *Armillaria* treatments (10^7^). Detected sesquiterpenes in *Armillaria* treatment (1,3,4,5,6,7-hexahydro-2,5,5,-trimethyl-2*H*-2,4a-ethanonaphthalene, β-barbatene, (*Z*)-α-bisabolene and δ-cadinene) were likely to have been emitted by the fungi themselves. Possible presence of 1,3,4,5,6,7-hexahydro-2,5,5-trimethyl-2*H*-2,4a-ethanonaphthalene and 4,4-dimethylpent-2-enal, common to both *A. gallica* and *A. ostoyae*, suggest intrageneric similarities in VOC emission for the *Armillaria* genus. Furthermore, the detection of β-barbatene in both *A. ostoyae* (a basidiomycete) and *F. poae* (an ascomycete) suggested possible similarities in VOC emission ranges of two different fungal divisions. β-barbatene is known to have been highlighted among volatiles of several Ascomycota (*Fusarium verticillioides*) and Basidiomycota (*Fomitopsis pinicola*, *Piptoporus betulinus* and *Trametes suaveolens*) [56]. In addition to the widespread 1-octen-3-ol and its C8 derivatives (such as 2-octen-1-al), fungi frequently emit mixtures of non-ramified, saturated or unsaturated, alcohols, aldehydes, esters and ketones, as well as various ramified methylated molecules [57]. Even though fungi are known to produce large amounts of terpenes, these molecules are found in some bacteria as well, particularly actinomycetes [58,59]. Some of these compounds directly act on the emitting organism’s close environment whereas others serve as intermediates in mycotoxins or other bioactive molecules biosynthetic pathways [60].

Detection of root rot by *Armillaria* species is currently very important in the green areas in contemporary cities. Harsh urban conditions such as soil compaction cause tree weakness and mortality of roots being covered by asphalt or bricks limiting access to oxygen and water. Most electronic devices such as resistographs or sound tomographs (PICUS) are designed for checking rot in tree trunks. There is no device for root examination without causing significant damage to plant tissues. In such circumstances the e-nose could address the issue of evaluating hazardous old trees and for the risk of dropping limbs or falling over, especially along streets where they might be poorly anchored.

#### 3.2.3. *Fusarium* Species

The genus *Fusarium* is one of the most dangerous to germinating seeds, causing damping-off seedlings. There is also a significant menace of introducing the quarantine pathogen *F. circinatum* from southern Europe, so early warnings about their unintentional entrance and establishment are needed.

##### *Fusarium avenaceum* 

The higher observed abundance magnitude (10^8^) in the *F. poae* chromatogram compared to the other studied *Fusarium* species (usually at 10^7^) was consistent with e-nose results indicating that this species was linked to a stronger sensor signal. The fact that longifolene was detected in *F. avenaceum* as well as in *F. poae* points towards possible intrageneric commonality in sesquiterpene emission. Furan and its derivatives (i.e., 2,4-dimethylfuran, but also 2-methylfuran, 3-methylfuran, 2,5-dimethylfuran, 2,3,5-trimethylfuran, 2-ethyl-5-methylfuran, etc.) are found in volatiles emitted by many fungal species. Moreover, the compound 1-octen-3-ol and its C8 co-metabolites can often be found together with 2-pentylfuran, which suggests a common biosynthetic pathway for these molecules. Other furan-derived compounds such as for instance 2-acetylfuran, 2-furanmethanol, 2-(methoxymethyl)furan and furfural are frequent as well [61]. β-bazzanene is, like trichodiene, an important precursor of various sesquiterpenoids in fungal metabolism [62]. Trichothecene mycotoxins for instance include powerful inhibiting compounds of eukaryotic protein biosynthesis [63].

##### *Fusarium culmorum* 

Production of ethyl acetate is common in yeasts, but has been observed in filamentous fungi such as *Ceratocystis fagacearum* [64]. The *F. culmorum* flask atmosphere appeared to contain several sesquiterpenes, as did all fungi (except *R. solani*) that were investigated in this study. These molecules most likely did come from the fungi and were not emitted by inert bodies of the sample (flask or medium).

##### *Fusarium oxysporum* 

Benzaldehyde, which was found in *F. oxysporum* and in the PDA control, may have been produced by the medium rather than the fungus itself. However, this molecule could theoretically have been emitted by the fungus, since benzene alkylated derivatives have been found to be produced by several fungal genera such as *Fusarium*, *Muscodor*, *Penicillium* and *Trichoderma* [57]. Whether these compounds fulfill any biological role or are just metabolic waste products still needs clarification. Benzaldehyde itself counts as one of the most widespread benzene derivatives among fungi (especially in the genera *Aspergillus*, *Botrytis*, *Fomes*, *Fusarium*, *Penicillium* and *Pleurotus*). Benzyl alcohol, methyl benzoate, ethyl benzoate and 4-methylbenzaldehyde are less common. Emission of benzothiazole was witnessed in *Aspergillus* and *Trichoderma* genera, even though pyrazines remain the most important group of nitrogenous fungal VOCs. Historically, the first aromatic fungal VOCs were identified in the middle of the 20th century in odoriferous decomposing wood. These molecules were thought to be synthetized from tyrosine and phenylalanine aromatic amino-acids or simply result from lignin degradation. The detected 3-chloro-4-methoxybenzaldehyde could have theoretically been released by the fungus as well, even though paraffine-derived chloroalkanes often serve in industry as ingredients for dyes and paste manufacturing. Indeed, chlorine aromatic compounds such as 4-chloro-1,2-dimethoxybenzene and 1,5-dichloro-2,3-dimethoxybenzene have been detected in the genus *Geniculosporium* [65]. Moreover, 3-chloro-4-methoxybenzaldehyde and 1,5-dichloro-2,3-dimethoxybenzene were identified in *Anthracophyllum discolor*, along with 3,5-dichloro-4-methoxybenzaldehyde [66]. The latter two molecules were furthermore observed in *Bjerkandera adusta*, together with their alcohol derivatives. Such compounds can be synthetized by fungi from the growth medium even if its chlorine concentration is very low. Analogous brominated compounds may be produced the same way, whereas iodine specks in growth medium can lead to diiodomethane or even chloroiodomethane biosynthesis according to Spinnler et al. [67].

##### *Fusarium poae* 

We found Matsutake alcohol, chemically called 1-octen-3-ol, which is a fatty acid characteristic for fungi widespread in the form of its R-enantiomer [68,69]. The R-enantiomer releases a fruity odor reminiscent of mushrooms, whereas the L-enantiomer is associated with grassy smell [70]. Initially detected in other fungi which were not tested in our experiment, such as *Tricholoma matsutake*, the alcohol has since then been identified in a wide range of fungal genera such as *Agaricus*, *Aspergillus*, *Boletus*, *Fistulina*, *Fomes*, *Phomopsis*, *Lentinus*, *Penicillium*, *Pleurotus*, *Tuber* and *Verticillium* but also in cultivated *Fusarium* and *Trichoderma* [57]. The compound 1-octen-3-ol is often found together with some of its C8 co-metabolites: octan-3-ol, octan-3-one, oct-1-en-3-one, octan-1-ol, oct-2-en-1-ol, octanal, trans-oct-2-enal, oct-1-ene and 1,3-octadiene. Some 1-octen-3-ol, octan-1-ol or octan-3-ol ester derivatives can be present in the mixture as well, in variable amounts. Linoleic acid is considered to serve as primary substrate for matsutake alcohol production, even though the exact biosynthetic pathway remains unclear. Production intensifies when the fruiting body is wounded, which could correspond to a defensive strategy of the fungus [71]. This confirmation of its presence in fungi which can help distinguish them from oomycetes is a significant finding for this study. However, it is worth speculating why 1-octen-3-ol is synthesized. First of all, it probably stops growth of some competing fungi, and indeed it has been found to inhibit conidial germination of several fungal species, including *Aspergillus nidulans* [72], *Lecanicillium fungicola* [73] and *Penicillium paneum* [74]. On the other hand, it was able to induce *Trichoderma atroviride* conidia germination [75]. This suggests 1-octen-3-ol acts as a hormone influencing fungal development. Its exact role however remains to be clarified. In our experiment we did use a mixture of organisms, but this should be done in future tests since there are reports that interactions between fungi are very important. Very often the volatile compounds are produced to inhibit or stimulate growth. Diverse species belonging to Ascomycota and Basidiomycota phyla, such as *Trichoderma harzianum*, were shown to produce bisabolene [76,77]. We found it from tested *Armillaria* species. Similarly, several longiborneol sesquiterpene derivatives showing antifungal properties were observed in *Fusarium* spp. (for instance in *F. culmorum* and *F. graminearum*) [78,79]. It was also observed in our study.

#### 3.2.4. *Phytophthora* Species

These organisms are considered to be serious primary pathogens of plants (including many forest tree species) and have often been unintentionally introduced to Europe from Asia. The observed increase of international trade of potted plants (e.g., bonsai) and seeds poses a new risk of establishment of alien, invasive species in forest stands. Also tourism, globalization, and quicker vessels with larger cargos accelerate this phenomenon. Therefore, an early detection of this group of organisms is of special importance.

##### *Phytophthora cactorum* 

Acetone is produced by *Clostridium acetobutylicum* and is frequently used as solvent by industry and research. Another compound—dimethyl disulphide—identified in our study is one of the most common sulfur molecules produced by bacteria, together with dimethyl sulphide, dimethyl trisulphide and S-methyl methanethiosulfonate [58]. Generally, bacteria are known to be more abundant emitters of sulfur compounds than fungi, although these latter may produce such substances as well. Less frequent in fungi, these latter compounds have been observed in several *Fusarium* and *Penicillium* species [57], but in our case were found in *P. cactorum* and *T. asperellum*. Dimethyl disulphide and other compounds such as benzonitrile, trimethyl disulphide or S-methyl thioacetate are volatiles with important antifungal effects [2]. On the other hand, *Aspergillus* and *Trichoderma* genera were shown to emit benzothiazole, which contains both sulfur and nitrogen. Moreover, cheese *Penicillium* strains can produce sulfur dioxide [80]. According to our results, oomycetes seem to be able to release sulfur compounds as well. Emission of 1-hexanol was witnessed in several fungal species such as *Aspergillus flavus* or *Fusarium fujikuroi*, which probably synthesize it as well as other related compounds through fatty acid degradation [81,82].

##### *Phytophthora cinnamomi* 

*P. cinnamomi* was rather poor in terms of VOC diversity. 4-ethyl-2-methoxyphenol is commonly found in beer and wine due to fermentation processes by yeasts from the genus *Brettanomyces* [83,84], which are fungi rather than oomycetes. 2-ethyl-1-hexanol was previously found in the PDA control, which makes it difficult to guess whether this compound was really emitted by the sample itself or originated from inert sample components. Some of the detected VOCs seemed to be common to *P. cinnamomi* and other species, including fungi.

##### *Phytophthora plurivora* 

Acetoin is an important metabolite (mainly carbon storage, physiological acidification avoidance and NAD/NADH ratio regulation), widespread in nature, emitted by many micro-organisms as soon as these have access to a degradable carbon source [85]. Bach et al. [86] have shown production of 4-hydroxybutanoic acid by *Saccharomyces cerevisiae*. In our experiment, substances such as acetoin and 4-hydroxybutanoic acid, α-Pinene, and Δ-3-Carene were found only in *P. plurivora* samples.

##### *Phytophthora ramorum* 

The peak abundances were higher in the *P. ramorum* chromatogram than for other *Phytophthora* species, in agreement with e-nose analysis results. 3-methyl-butanol and similar alcohols are often emitted by endophytic fungi belonging to *Phoma* and *Phomopsis* genera, which are able to break down cellulose [87,88]. More generally, 3-methyl-butanol is frequently observed in fungi, and is probably synthesized from leucine. *Ceratocystis paradoxa* was the first fungal strain to be witnessed producing 3-methyl-butanol in its volatile mixture [89]. 2-phenylethanol, widespread among microorganisms, is an intermediate of l-phenylalanine aromatic amino acid metabolism. Thus, numerous yeast species (including *Candida albicans* and *Saccharomyces cerevisiae*) synthesize it as an antibiotic [90,91]. Rapior et al. [92] observed the presence of 2-phenylethanol in VOC mixtures emitted by several Basidiomycetes species. It is common in many fungi genera: *Aspergillus, Chaetomium, Fusarium, Hypoxylon, Lasiodiploida, Penicillium, Polyporus, Trichoderma, Tuber*, etc. [57]. Production of 2-phenyethanol using microorganisms could serve the industry to avoid artificial synthesis with implied purification steps, which makes the whole process very expensive [93,94]. *P. ramorum* in Europe causes sudden larch decline in western Great Britain, so its further spread in Europe should be closely monitored.

#### 3.2.5. *Rhizoctonia solani* 

There appeared to be a big difference between the low abundance and diversity of the *R. solani* chromatogram and the strong induced reaction from e-nose sensors (Figure 1m). One possible hypothesis is that this species may emit many volatile inorganic compounds (VICs) detected by the e-nose sensors but not extracted by the SPME fiber. Alternatively, molecules released by *R. solani* may simply have intrinsically higher stimulation of the e-nose, meaning they “smell” stronger to the sensors. Drilling and Dettner [95] witnessed emission of 3-octanol by *Trametes versicolor*, independent of 1-octen-3-ol emission.

#### 3.2.6. *Trichoderma asperellum* 

Consistent with e-nose results (Figure 1), the *T. asperellum* chromatogram showed a high abundance magnitude (almost 10^8^). Literature indicates that trans-dauca-4(11),8-diene was spotted in *Omphalotus olearius*, along with α-barbatene, β-barbatene and γ-cadinene [96]. It was also observed in *Schizophyllum commune* [97] and in *Bjerkandera adusta* [98]. Isodaucene was detected in several fungal species, such as *Tricholomopsis rutilans* [99] and *Aspergillus fischeri* [100]. Cedrene has been isolated in several fungi such as *Corynespora cassiicola* and *Beauveria sulfurescens*, but also in soil bacteria *Rhodococcus rhodochrous* [101]. VOCs emitted by the *Trichoderma* genus have been shown to play the role of signal compounds for communication between colonies as well as for growth regulation. For instance, molecules such as 3-octanol, 3-octanone and 1-octen-3-ol emitted by conidia of *Trichoderma* cultures induced conidiation of other colonies from the same genus [75].

#### 3.2.7. Strategies for Improved Detection

The compounds mentioned above were often shared among organisms but some of them appeared only once suggesting specificity. However, our list of organisms was very limited so if we were to confirm specificity of some compounds, we would need to do so in the future broader screening of organisms. We could apply precise methods such as GC/MS is to identify chemical compounds or to use e-nose sensors. Both approaches have their advantages and disadvantages. Semiconductor sensors used in the e-noses usually show a wide sensitivity range and a non-linear response with respect to gas concentration. Such sensors provide a quick response, involve a simple (thus inexpensive) circuit design and have a long lifetime. The output signal of a given sensor arises from the superposition of individual effects of every component of the gaseous mixture. Because individual sensors (e.g., TGS—Figaro Gas Sensors) show a lack of selectivity, combining several of them with different cross sensitivities allowed improved performance. On the other hand, too many sensors taken together will increase measurement noise.

In order to know which sensor is suitable for a particular chemical compound or their group, other techniques should be applied e.g., multidimensional data analysis tries to highlight odor patterns that can be used to characterize gaseous mixtures based on identification, comparison and classification principles. First studies investigating microbial volatiles involved steam distillation and liquid-liquid extraction followed by compounds concentration and identification [7]. Since the gas chromatography (GC) is becoming an affordable and reliable detection method, scientists have merged the separation, identification and quantification steps into one single analytic process, GC/MS. We also used this technique in our experiments because of its high sensitivity and strong discriminatory ability. GC/MS is today the most used analytical tool for fungal VOC identification [102,103,104] and is easily coupled to solid-phase microextraction (SPME) techniques for the VOC extraction and concentration, which allows easy progression to environmental sampling and subsequent laboratory analysis [100]. However, it should be noted that results observed depend on both the nature of the fiber used and the extraction method [105]. Artifacts can sometimes appear due to solid sorbents serving for headspace analysis [7]. The atmospheric water content of the sample can also bias GC/MS data since fungal VOC formation is easier in humid ambient air. Furthermore, such physical analyses require some time to be implemented [106], even though GC/MS remains a relatively quick way of analyzing VOC mixtures, which can be furthermore automated for real time profiling of compounds emitted from living fungi [107,108,109].

Alternatively, other analytical methods may be used for VOC characterization. For example, the simultaneous distillation extraction (SDE) method relies on simultaneously occurring vapor distillation and solvent extraction [105]. It is especially applied for the extraction of high boiling volatiles with the flaw of potentially making false features appear because of the longstanding effect of heat. The selected-ion flow-tube mass spectrometry (SIFT-MS) is a technique allowing for a real time measurements of VOC concentration in a sample’s atmosphere, with a high degree of sensitivity (up to a few ppb) [110]. It allows quick characterization of a gaseous mixture composed of a wide range of molecules. The proton-transfer-reaction mass spectrometry (PTR-MS) involves the VOC ionization using H_3_O^+^ primary ions, which results in the MH^+^ ions production (where M stands for a neutral organic molecule). These are in turn detected by the means of a quadrupole mass spectrometer [111]. This method provides a sensitivity which is comparable to GC/MS, however in contrast to GC/MS affords the robust analysis of a sample without any pre-processing or pre-concentration [112,113].

Based on what was found in our samples and the above literature review, we hypothesize that some substances (listed in Table 1) are also specific to particular organisms. Even if we do not know their putative functions, we believe that they can be used to discriminate between individual fungal and oomycetes species or at least between genera. The only problem could be obtaining sensors for reasonable prices for the most specific compounds for each organism in order to construct new models of e-nose.

Alternatively, further developments may go in the other direction of determining not just single or a few specific compounds, but to train neural networks (artificial intelligence) how to recognize differences among samples e.g., like pictures. In such a case we are not concerned about the chemistry of VOCs (its content), but the focus is to point out differences between organisms. For this purpose, much empirical data will be needed, as well as technical details that need to be worked out such as timing of each measurement, its temperature, and the required humidity. Furthermore, specific cultural conditions for microbial growth, may also affect the quality and quantity of emitted VOCs.

## 4. Materials and Methods

### 4.1. Sample Material Preparation

In total, 144 jars were tested by e-nose containing pure cultures of strains (one to four-week-old fungi and oomycetes prepared in three repetitions). They consisted of 12 different species important in forestry, and were obtained from stocks kept in the laboratory of the Forest Research Institute (IBL) (Table 2). In addition, three copies of aforementioned organisms were grown for VOC analysis by the GC-SM method.

Transfers of mycelia were made from stored pure colonies onto PDA (20 g glucose + 15 g agar + 4 g potato extract dissolved in 1 L distilled water) using powder purchased at BTL Ltd. (Łódź, Poland), and cultured at room temperature. The investigated microorganisms were raised in specially constructed 300 mL glass flasks or 40 mL glass vials. Each flask was fitted with a 66-mm-diameter lid made of galvanized steel with a silicone ring on the underside to prevent air exchange. Each lid was made a 9-mm-diameter hole covered with sterilization tape to allow subsequent collection of gaseous samples by introducing the e-nose inlet or a SPME syringe inside the flask through the hole, thus removing or piercing the closing tape. Similarly, all vials were sealed by a polypropylene cork provided with a polyisobutylene-polytetrafluoroethylene (PIB-PTFE) septum contiguous to its bottom side. As previously, the cork was perforated so that a SPME syringe could later on be introduced in the vial atmosphere by only piercing the underlying septum. All the dishes were distributed in the prepared PDA medium, sealed and autoclaved at 117 °C and 0.08 MPa for 20 min, prior to microorganisms’ inoculation under sterile conditions. Subsequently, in every week (up to four weeks), each of the three jars for every organism was measured with the e-nose devices.

### 4.2. E-nose Device and Measurements

The e-nose was developed by the Warsaw University of Technology (WUT) based on the e-nose sensing device, which has been used in previous research [48,49] and was made of two homologous sets of gas sensors produced by Figaro Engineering Inc. (Osaka, Japan), which include a semiconductor tin oxide layer and arrays made of six different heated metal oxide gas sensors types (Figaro Engineering Inc. Osaka, Japan). Four sensor types were duplicated inside each array (Appendix A). Duplicated sensors of the same array had a slightly different loading setting, so that they don’t show exactly the same sensitivity spectrum: their output signals were therefore somewhat different, and for instance the 26xx sensor series family was chosen for its small size and high stability of operation. The sensors were not wired in series but were connected to different analog inputs of the data acquisition unit, one by one, and their loading was tuned in a clean air environment using a potentiometer. Moreover, two sensors measuring, respectively, relative humidity (HIH-3610-002 from Honeywell, Morristowne, NJ, USA), and temperature (LM35DH from Texas Instruments, Dallas, TX, USA) were included in both arrays, since those parameters impact sensor sensitivity (Appendix A).

Each of the two independent sensor arrays were seated on a custom designed printed circuit board, and installed in a specially designed optimized test chamber consisting of an aluminum black cylinder (approximately 500 cm^3^ volume) with both extremities connected to a socket outlet consisting of an 8.5 mm external diameter rubber tube (Figure 3a,b). Both cylindrical channels were standing nearby, in the same ambient environment. The sensor electrical conductivity (resistance) changed according to the concentration of molecules belonging to their respective sensitivity spectra in the test chamber and was converted by a simple electrical circuit into an output tension voltage signal. One set formed the measuring device and was put in contact with the sample atmosphere to be analyzed, whereas the other kept sampling ambient air thus providing a reference signal (Figure 3c). Both sets of sensors delivered their own real time signal when the measurement was launched. However, only the difference was continuously saved and transferred to the computer through a serial communication interface consisting of two 8-Channel Analog Input Modules Rev. D1 type ADAM-4017 built by Advantech (Taipei, Taiwan).

The deduction between analog signals from both sensor arrays was done by a differential amplifier, whereupon an A/D converter turned the signal into digital format. Differential profiles obtained this way were made of the weighted sum of sensor reaction to the gaseous mixture respective components. This differential functioning removed the need to perform a systematic calibration of the reference signal prior to each measurement at changing conditions of ambient air. It also greatly reduced the influence of natural drift due to ageing of the metal oxide sensors as well as the common distortion effect linked with variations of pressure or temperature in the test chamber. Furthermore, the impact of potential intrinsic differences between two homologous sensors or operation errors was in this way minimized. Finally, it increased the sensitivity towards low concentration compounds.

An inducting pump was used to set up the air intake with a laminar and almost one-dimensional flow entering the channel through the socket outlet. It was kept constant by a flowmeter set to 1 L·min^−1^. Twelve samples of each treatment were tested by introducing the rubber tube inlet in the sample’s atmosphere through the hole on the flask’s lid, from which the sterilization tape had been previously removed. The samples were tested at a rate of three per week, up to four weeks after inoculation. Each sample was analyzed only once. The volumes of the test chamber as well as the volume of the sample atmosphere were kept constant during the measurement process. The measurement window duration was arbitrarily set to 300 sec with a resistance sampling rate of 60 times per min. During measurements, which were performed in a dynamic on-line mode, the sensor temperature would fluctuate in a range of 27 to 35 °C, with an almost constant sensor chamber RH of 14–15%.

A washing interval was implemented between every two successive measurements during which the system was let running on itself for typically 10 min so that the measuring array could desaturate from the preceding sample’s atmosphere molecules. However, implementing a more accurate procedure would have meant injecting synthetic air into both test chambers for 15 min. Furthermore, a blank analysis (without submitting any sample to the sensing device) was carried out every time before each new measurement series. Due to imperfect coordination in the setting of homologous sensors of the two sensing arrays, a very low intensity baseline signal would appear which was in turn removed from the pattern yielded from each subsequent analysis.

The experimental data set for each sample consisted of a matrix including 300 vectors in a twelve-dimensional space. Only measured signal values from the final period of the measuring window were used to obtain diagnostic features, so that the signal could settle down and reach a steady state before being considered. Thus, the initial rapidly increasing stage of the sensor signals derived from the transient effect was not taken into account for descriptive features generation, which was done by averaging the second half (last 150 measurements) of every temporal series of sensor resistances *R*(_j_), referring to each *j*-th sensor of the array:r_(*j*)_ = R_(*j*)_ − R_0(*j*)_(1)
with *R*_(*j*)_ representing the averaged measured resistance of the *j*-th sensor of the array and *R*_0(j)_ standing for the averaged baseline value of resistance measured during the blank analysis, both calculated for the *j*-th sensor of the array based on the last 150 measurements of the corresponding temporal series. Thus, each analyzed sample would yield a final diagnostic feature consisting of a 12-dimensional vector (r1, …, r12) quantifying the reaction of e-nose sensors exposed to the tested material. Overall, twelve such vectors were obtained per treatment, corresponding to the 12 repetitions performed. Calculating the mean of these 12 diagnostic features allowed us to obtain a global averaged signal vector for each treatment, which could be pictured as a diagram. This final 12-dimensional vector was considered as representing the mean e-nose sensor reaction to the treatment, thus providing an assessment of the tested material odor intensity [22,110].

### 4.3. SPME-GC/MS Analysis

To validate the previously described method of detection with e-nose, VOCs emitted by tested microorganisms were investigated using headspace SPME-GC/MS in two different laboratories at the BUT and at the Institute of Physical Chemistry of the Polish Academy of Sciences in Warsaw (IPC-PAS). A two way approach was performed in order to better detect the VOCs emitted by tested microorganisms, i.e., the first one based on the 85 μM PDMS/Carboxen fiber (in the BUT laboratory), and the second one based on 100 μM diameter PDMS fiber in the IPC-PAN laboratory. All SPME-GC/MS measurements were for the first time applied to the fungal species, and hence the detailed explanation.

In the first approach, the samples were analyzed with a SPME syringe including a 85 μM diameter PDMS/Carboxen fiber (Supelco, Bellefonte, PA, USA). The fiber was heated to 250 °C for 1–2 h after the purchase to remove any potential contaminant adsorbed on the coating. Similarly, this process was repeated for 5–10 min before testing each new set of samples. The fiber was introduced in the flask through the lid hole and placed in an incubator under 40 °C where a 30 min extraction took place. The fiber was then introduced for 10 min in the chromatograph injection device, where a temperature of 250 °C would allow the compounds to be desorbed. The injection was done in a splitless mode. An Agilent 7890A gas chromatograph including a 30 m × 0.25 mm × 0.25 μM HP-5MS semipolar capillary column connected to an Agilent 5975C mass spectrometer (Agilent Technologies, Santa Clara, CA, USA) was used for GC/MS analysis, which lasted 43 min overall. From an initial temperature of 35 °C, the oven was heated at a rate of 5 °C min^−1^ up to 250 °C. The electron ionization potential was set at 70 eV and the electron ionization (EI) source worked at the temperature of 230 °C. The temperature of the quadrupole analyzer was 150 °C. Helium circulating through the column at a steady flow of 1 mL·min^−1^ served as a carrying gas. The spectrometer was working in a full scan mode over a 29–600 mass range. Recorded EI mass spectra were compared against the NIST Mass Spectral Database and Willey libraries for analyte identification. For some unknowns, the identification was supported by an in-house constructed library comprising the EI mass spectra for available standards. In addition to the MS spectra, also RI and authentic standards were used for identification of the compounds.

The IPC-PAS laboratory pursued analyses of the samples with a SPME syringe equipped with a 100 μM diameter PDMS fiber (Supelco, Bellefonte, PA, USA). The fiber was preconditioned before each analysis in a GC/MS injector where it was heated at 250 °C during 30 min. Then, it was introduced in a vial by piercing a cork’s underlying septum to perform the extraction, which took place at room temperature and lasted 60 min. Subsequently, a 3 min desorption was performed at 250 °C in a GC injector. The injection was done in a splitless mode. The chromatographic analysis lasted overall 28 min and was performed in a Thermo Trace 1300 gas chromatograph equipped in a 30 m × 0.25 mm × 0.25 μM Rtx-5MS semipolar capillary column coupled with a Thermo ITQ 700 mass spectrometer (Thermo Scientific, Waltham, MN, USA). The initial oven temperature was kept isocratic at 100 °C for 5 min, and then increased at a rate of 10 °C min^−1^ to reach a peak temperature of 280 °C at which it was kept for 5 min. The MS detector was equipped with a 70 eV electron ionization (EI) source. The temperature of the EI ion source was 250 °C, while the quadrupole ion trap analyzer was 250 °C. The carrying gas was helium circulating through a column at a steady flow of 1 mL·min^−1^. The spectrometer was working in a full scan mode for a 50–650 mass range. The EI mass spectra obtained were matched with these from the NIST and Willey Mass Spectral Databases for the molecular elucidation.

In both analyses, a ranking showing a list of candidates with the best matches and corresponding estimated confidence rates was generated by the analyzing software. For GC/MS analyses, we used the following software: MSD ChemStation E.02.02.1431, Agilent Technologies and NIST MS Search 2.0. The default analytical software obtained with each instrument was used for calculations for each spotted compound [22,114].

### 4.4. Statistical Analysis

The variability among signals obtained from twelve sensors of the e-nose device was computed by principal component analysis (PCA) in “R” software [115]. PCA analyses as well as the biplot were created with fviz_pca_biplot functions from the “FactoMineR” 1.41 package [116]. The variables with the strongest impact on the distribution of the microorganisms along the principal components were identified on the basis of Pearson correlation coefficients.

To group microorganisms according to their signal similarity among twelve sensors, a hierarchical clustering using Euclidean distance (root sum-of-squares of differences) as the similarity measure and Ward [117] clustering method with the criterion proposed by Murtagh and Legendre [118] were applied. Four different clustering methods, single and complete linkage, the unweighted pair group method with arithmetic mean (UPGMA), and Ward’s method were tested according to the clustering structure of the dataset [119]. Ward’s method was chosen because it expressed the highest value of the agglomerative coefficient. Hierarchical clustering was performed using the Cluster 2.0.7-1 package [120]. Final grouping of microorganisms according to their locations on the PCA biplot were determined visually in Figure 2.

## 5. Conclusions

For the first time, we demonstrated that the e-nose apparatus was able to distinguish between VOCs emitted by the investigated fungi or oomycetes, but further technical developments are still needed for its practical use in the forestry sector in the field or in practice, i.e., nurseries, plantations, stands, or in quarantine laboratories.

Results of testing with the e-nose prototype showed that certain fungal species, such as *F. poae*, *R. solani* and *T. asperellum*, were the most odoriferous among the studied organisms, and gave rise to the strongest signals. In our in vitro study, *fusaria* (*F. avenaceum*, *F. culmorum* and *F. oxysporum*) and an oomycete (*P. ramorum*) generated moderately intense signals as detected by the e-nose. Two tested *Armillaria* species, *A. gallica*, *A. ostoyae*, and three oomycetes, *P. cactorum*, *P. cinnamomi* and *P. plurivora*, generated specific odors detected by the e-nose sensors, e.g., α-Pinene and Δ-3-Carene for *P. plurivora*. The Principal Component Analysis plot revealed that our system of e-nose detection could discriminate between the odors emitted by *P. ramorum*, *F. poae*, *R. solani* and *T. asperellum*, making this device suitable for practical use in laboratory situations, at least for the species tested.

Identification of VOCs detected by e-nose was revealed when using two carbon fibers SPMEs. The tested PDMS/Carboxen fiber was more efficient for fungal detection compared to PDMS fiber. The majority of VOCs detected were specific compounds to the genus or species level, due to a complex mixture of (un)saturated (non-)ramified hydrocarbons and their oxygenated derivatives (aldehydes, alcohols, esters). We also noticed that all tested fungal species released sesquiterpenes in variable amounts, apart from *R. solani*. All tested *Phytophthora* strains emitted none of these compounds. Therefore, in the future, development of future stage e-nose apparatus will rely on unique volatile compounds identified and specific to investigated microorganisms.

This research should be of special interest for quarantine organizations (e.g., National Plant Protection Organizations or international ones like EPPO) dealing with alien invasive species such as *P. ramorum*. Further studies need to be carried out with protocol designs allowing control of intraspecific and temporal variability of VOC mixture profiles dependent on environmental conditions, especially in forest nurseries.

## Figures and Tables

**Figure 1 molecules-25-05749-f001:**
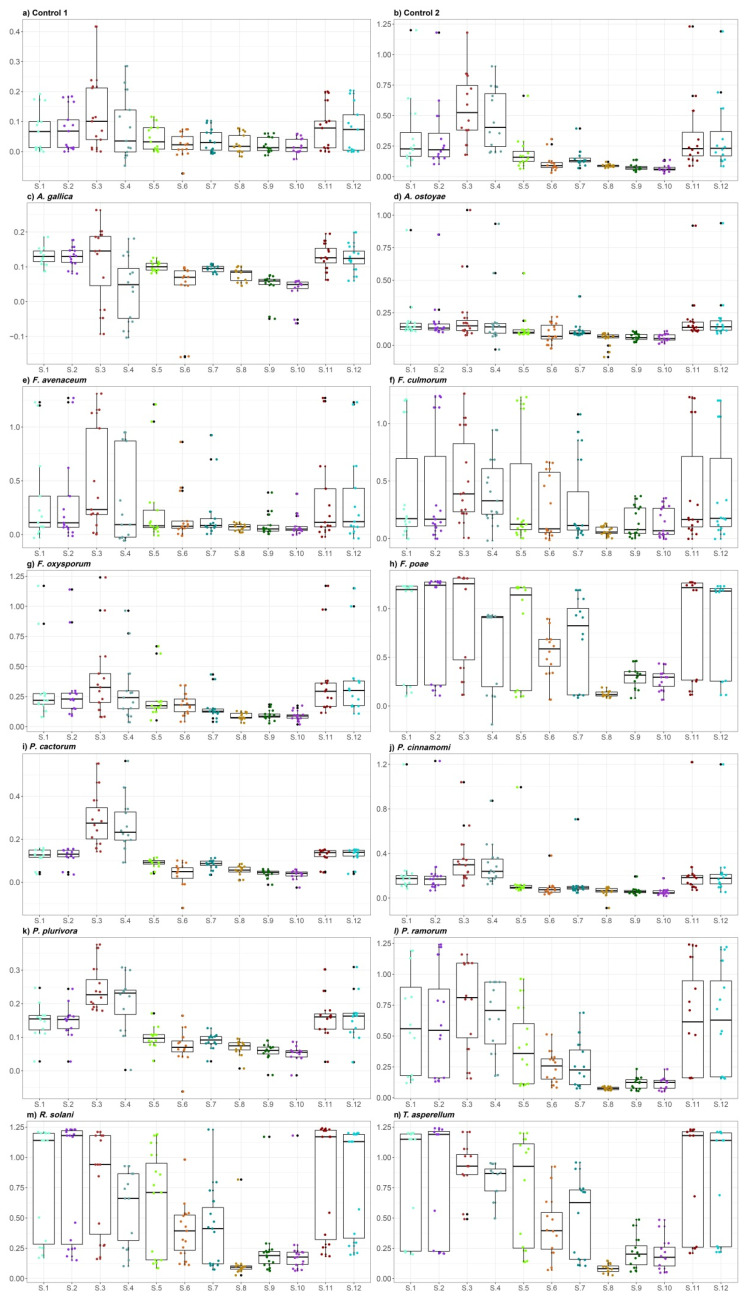
Distribution of the overall averaged values of twelve differential sensor original signals for the respective 14 treatments; e.g., (**a**) control 1 (empty flask); (**b**) control 2 (PDA medium); (**c**) *A. gallica*; (**d**) *A. ostoyae*; (**e**) *F. avenaceum*; (**f**) *F. culmorum*; (**g**) *F. oxysporum*; (**h**) *F. poae*; (**i**) *P. cactorum*; (**j**) *P. cinnamomi*; (**k**) *P. plurivora*; (**l**) *P. ramorum*; (**m**) *R. solani*; (**n**) *T. asperellum*.

**Figure 2 molecules-25-05749-f002:**
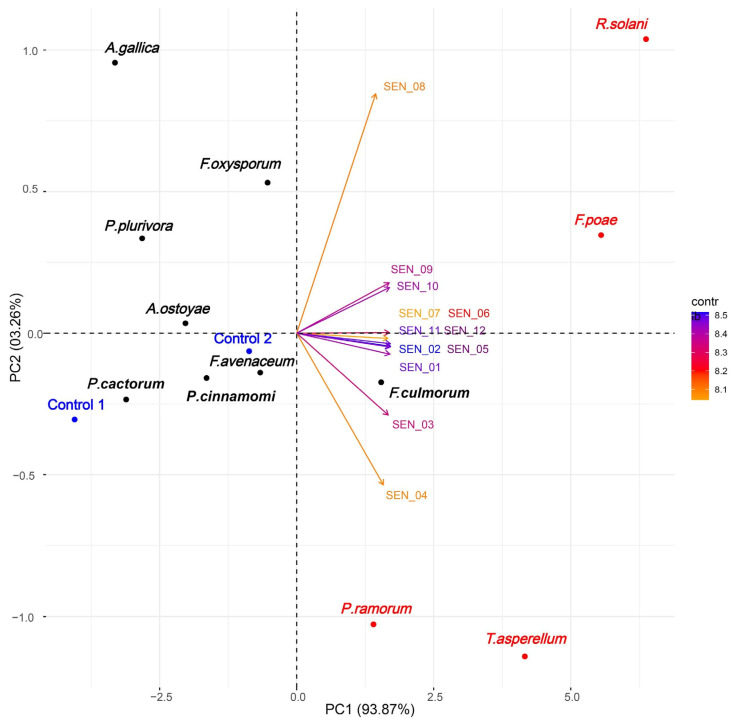
Variability of twelve differential sensor signals assigned to the twelve investigated microorganisms. Vectors were colored according to their contribution to total variance: orange—low, blue—high. Blue dots represent controls: 1—empty flask, and 2—PDA medium; red dots—microorganisms with the most pronounced eigenvalues, black dots—microorganisms with less pronounced eigenvalues (cf. Appendix A, Appendix A).

**Figure 3 molecules-25-05749-f003:**
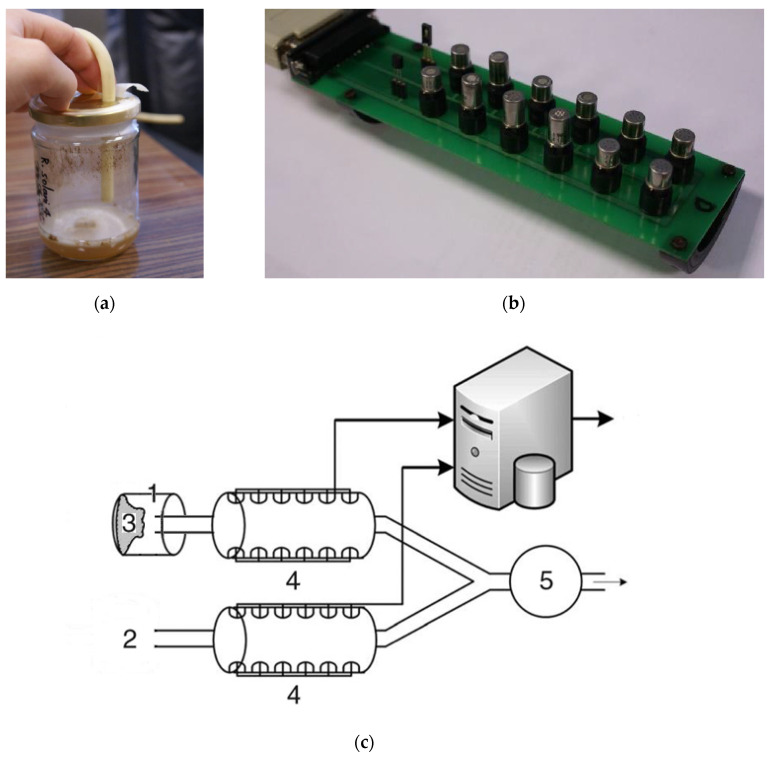
Representative view of the differential electronic nose apparatus. (**a**) Introduction of the air intake tube inside the sample; (**b**) Twelve chemical sensors in an array chamber; (**c**) General scheme of the e-nose principle of measurement: (1) Sample flask; (2) Ambient air; (3) Tested material; (4) Sensor chambers; (5) Inducting pump and flow-meter. Adapted from Brudzewski et al. [22], Osowski et al. [114].

**Table 1 molecules-25-05749-t001:** The VOC emission pattern detected using a SPME-GC/MS method in the headspace atmosphere of samples tested at BUT. Similarity indices (SI) were obtained by comparing a profile of the EI mass spectrum of a given VOC compound with the reference one in NIST Mass Spectral Database library; these values represent the confidence in the molecular identification.

t_R_ ^1^ [min]	SI ^2^ %	RI_Exp_ ^3^	RI_Lit_ ^4^	Ref.	Peak Area per Treatment	Name of Compound ^5^
				(1) Empty Container		
7.19	65	902	901	[29]	17,674,234	Heptanal ^++a^
10.05	80	1003	998	[29]	40,444,566	Octanal ^++a^
10.82	28	1029	1027	[30]	21,189,032	2-Ethyl-hexanol ^a^
13.05	79	1104	1100	[29]	76,538,207	Nonanal ^++a^
15.98	53	1206	1201	[29]	14,513,924	Decanal ^++a^
18.06	−	1281	−	−	14,724,741	Unknown compound
18.77	−	1307	−	−	13,630,063	Unknown compound
21.21	32	1400	1400	[29]	6,427,563	Tetradecane ^++a^
21.44	58	1409	1408	[29]	17,444,416	Dodecanal ^++a^
24.07	78	1516	1514	[31]	47,388,559	Butylated Hydroxytoluene ^++a^
					(2) PDA Medium	
4.77	50	804	801	[29]	19,939,473	Hexanal ^+a^
5.48	57	834	828	[29]	11,223,384	Furfural ^+a^
7.18	31	902	901	[29]	39,366,443	Heptanal ^++a^
8.80	72	959	952	[29]	60,063,839	Benzaldehyde ^a^
9.74	87	992	984	[29]	31,027,584	2-Pentyl-furan ^b^
10.05	74	1003	998	[29]	11,751,703	Octanal ^++a^
10.81	44	1029	1027	[30]	32,462,061	2-Ethyl-hexanol ^a^
11.23	73	1043	1036	[29]	73,539,705	Benzeneacetaldehyde ^++a^
12.96	21	1101	1100	[29]	12,884,326	Undecane ^+a^
13.04	80	1104	1100	[29]	41,840,277	Nonanal ^++a^
15.48	−	1189	−	−	4,687,953	Unknown compound
15.83	30	1201	1200	[29]	4,893,511	Dodecane ^+a^
15.97	50	1206	1201	[29]	10,723,873	Decanal ^++a^
18.13	−	1284	−	−	5,876,115	Unknown compound
18.77	−	1307	−	−	4,815,975	Unknown compound
21.43	44	1409	1408	[29]	11,027,672	Dodecanal ^++a^
24.06	74	1515	1514	[31]	14,730,122	Butylated Hydroxytoluene ^++a^
					(3) *A. gallica*	
8.80	47	959	n/a^6^	−	300,374,347	(2*E*)-4,4-Dimethyl-2-pentenal ^c^
11.73	53	1060	1049	[29]	62,813,700	(*E*)-2-Octen-1-al ^b^
20.87	10	1387	n/a	-	611,543,126	1,3,4,5,6,7-Hexahydro-2,5,5-trimethyl-2H-2,4a-ethanonaphthalene ^c^
23.89	26	1506	1506	[29]	74,690,325	(*Z*)-α-Bisabolene ^+a^
					(4) *A. ostoyae*	
8.80	49	959	n/a	-	87,776,838	(2*E*)-4,4-Dimethyl-2-pentenal ^c^
11.74	35	1060	1049	[29]	20,166,139	(*E*)-2-Octen-1-al ^b^
18.49	21	1297	1293	[29]	8,070,650	2-Undecanone ^a^
20.83	43	1386	n/a	-	106,170,662	1,3,4,5,6,7-Hexahydro-2,5,5-trimethyl-2H-2,4a-ethanonaphthalene ^c^
22.49	59	1451	1440	[29]	43,908,793	β-Barbatene ^a^
24.42	34	1531	1522	[29]	24,134,794	δ-Cadinene ^+a^
					(5) *F. avenaceum*	
3.28	76	712	735	[32]	2,653,753	2,4-Dimethylfuran ^+b^
9.13	23	971	970	[33]	3,974,721	Mesitilene ^+a^
10.91	76	1032	1029	[29]	9,540,610	β-Phellandrene ^a^
21.59	35	1415	1407	[29]	11,961,247	Longifolene ^a^
24.46	73	1532	1519	[29]	3,401,805	β-Bazzanene ^+a^
				(6) *F. culmorum*		
1.62	77	<500	448	[34]	163,586,144	Ethanol ^a^
2.25	96	573	606	[29]	45,234,113	Ethyl acetate ^+a^
3.55	41	728	731	[29]	6,085,804	Isoamyl alcohol ^+a^
3.61	56	732	724	[29]	3,371,660	2-Methyl-butanol ^a^
10.91	18	1032	1029	[29]	177,256,946	β-Phellandrene ^a^
21.80	43	1424	1408	[29]	20,993,355	Acora-3,7(14)-diene ^+b^
22.23	man.^7^	1441	1432	[29]	138,760,275	β-Copaene ^+a^
					(7) *F. oxysporum*	
6.02	78	854	864	[35]	845,609	2-Furanmethanol ^+b^
7.96	59	930	927	[36]	2,621,568	Hexyl formate ^+a^
8.86	79	961	952	[29]	49,523,292	Benzaldehyde ^+a^
9.65	68	989	979	[29]	5,057,096	3-Octanone ^a^
10.27	71	1010	960	[29]	2,393,859	Isoamyl propionate ^+a^
10.90	54	1032	n/a	−	63,835,818	Heptyl formate ^+a^
21.98	33	1431	1419	[29]	4,969,490	β-Cedrene ^a^
22.49	90	1451	n/a	−	23,065,220	3-Chloro-4-methoxy-benzaldehyde ^c^
22.77	20	1463	n/a	−	8,382,097	2,4-Dichloro-3-methoxy-1-benzene carbonyl chloride ^+c^
23.34	93	1486	864	[35]	2,740,362	3,4-Dimethoxy-benzaldehyde ^+a^
					(8) *F. poae*	
9.43	81	981	974	[29]	93,704,125	1-Octen-3-ol ^a^
9.79	36	994	988	[29]	166,758,547	Myrcene ^+a^
13.39	59	1116	1122	[37]	100,643,564	1-Ethyl-4-methoxy-benzene ^b^
16.68	41	1231	1223	[29]	36,663,640	Citronellol ^+a^
20.10	28	1358	1350	[29]	106,101,600	α-Longipinene ^+a^
20.65	16	1379	1371	[29]	108,987,407	Longicyclene ^+a^
21.43	man.	1409	1400	[29]	248,606,376	β-Longipinene ^+b^
21.65	37	1418	1407	[29]	1,390,824,920	Longifolene ^a^
21.81	−	1427	−	−	482,683,621	Unknown sesquiterpene
21.91	22	1428	1419	[29]	195,664,192	β-Ylangene ^+a^
22.31	−	1444	−	−	954,381,082	Unknown sesquiterpene
22.55	55	1454	1440	[29]	129,567,010	β-Barbatene ^+a^
22.68	16	1459	1449	[29]	117,848,857	α-Himachalene ^+a^
22.75	50	1462	1454	[29]	143,907,373	(*E*)-β-Farnesene ^+a^
23.02	20	1473	1466	[29]	116,555,130	α-Acoradiene ^+a^
23.38	16	1487	1481	[29]	134,783,119	γ-Himachalene ^+a^
23.98	43	1512	n/a	−	44,779,678	8-Isopropenyl-1,5-dimethyl-1,5-cyclodecadiene ^+c^
24.05	17	1515	1505	[29]	51,262,732	β-Bisabolene ^+a^
25.03	−	1557	−	−	42,987,721	Unknown sesquiterpene
26.31	60	1612	1599	[29]	1,576,920,976	Longiborneol ^+a^
				(9) *P. cactorum*		
1.71	86	<500	500	[34]	5,827,788	Acetone ^a^
3.75	97	740	744	[38]	4,459,515	Dimethyl disulfide ^b^
4.12	−	763	−	−	4,321,348	Unknown compound
6.35	60	868	863	[29]	25,809,441	1-Hexanol ^+a^
9.15	58	971	959	[29]	5,111,166	1-Heptanol ^+a^
9.42	75	981	974	[29]	19,848,783	1-Octen-3-ol ^a^
9.65	76	989	979	[29]	12,627,315	3-Octanone ^a^
9.81	88	995	984	[29]	4,284,441	2-Pentyl-furan^a^
12.05	22	1071	1060	[29]	2,508,277	2-Octen-1-ol ^+a^
12.13	19	1073	1063	[29]	4,044,266	1-Octanol ^+a^
				(10) *P. cinnamomi*		
1.61	97	<500	448	[31]	9,483,339	Ethanol ^a^
9.43	56	981	974	[29]	2,215,655	1-Octen-3-ol ^a^
10.88	47	1031	1027	[30]	5,251,305	2-Ethyl-hexanol ^a^
18.13	85	1284	1282	[39]	1,001,882	4-Ethyl-2-methoxy-phenol ^+b^
				(11) *P. plurivora*		
1.71	84	<500	500	[34]	4,481,938	Acetone ^a^
3.20	64	707	711	[40]	2,268,583	Acetoin ^+a^
6.37	36	869	863	[29]	1,803,069	Hexanol ^+a^
7.51	76	914	933	[41]	3,627,373	4-Hydroxy-butanoic acid ^+b^
8.14	16	936	932	[29]	1,684,400	α-Pinene ^+a^
10.35	14	1013	1008	[29]	1,103,050	Δ-3-Carene ^+a^
				(12) *P. ramorum*		
1.16	98	<500	448	[34]	20,885,799	Ethanol ^a^
3.55	74	728	730	[42]	80,424,948	3-Methyl-butanol ^a^
3.61	74	732	724	[29]	26,123,097	2-Methyl-butanol ^a^
9.43	72	981	974	[29]	8,370,575	1-Octen-3-ol ^a^
9.65	70	989	979	[29]	11,144,929	3-Octanone ^a^
13.37	87	1116	1106	[29]	48,377,677	2-Phenylethanol ^+a^
					(13) *R. solani*	
1.71	87	<500	500	[34]	3,565,392	Acetone ^a^
9.42	74	981	974	[29]	24,589,405	1-Octen-3-ol ^a^
9.65	62	989	979	[29]	3,438,485	3-Octanone ^a^
9.90	75	998	988	[29]	6,372,403	3-Octanol ^+a^
15.56	−	1191	-	-	4,474,717	Unknown compound
				(14) *T. asperellum*		
1.71	82	<500	500	[34]	3,920,210	Acetone^a^
3.56	62	729	730	[41]	7,870,509	3-Methyl-butanol^a^
3.62	45	733	724	[29]	4,424,960	2-Methyl-butanol^a^
3.76	98	741	744	[38]	11,829,339	Dimethyl disulfide^b^
7.26	−	905	−	−	153,721,663	Unknown compound
13.01	−	1103	−	−	20,319,451	Unknown compound
13.38	85	1116	1106	[29]	37,177,146	2-Phenyethanol ^+a^
20.08	−	1357	−	−	70,272,034	Unknown sesquiterpene
20.91	44	1389	1380	[29]	229,806,310	Daucene ^+b^
21.81	man.	1424	1412	[29]	25,748,186	2-epi-β-Funebrene ^+b^
21.98	50	1431	1419	[29]	86,596,372	β-Cedrene^a^
23.93	28	1510	1500	[29]	118,470,969	Isodaucene ^+b^
24.27	21	1524	1513	[29]	52,189,482	γ-Cadinene ^+a^
24.63	40	1540	1530	[29]	194,592,295	Dauca-4(11)-8-diene ^+a^
25.49	−	1576	−	−	8,894,971	Unknown sesquiterpene
27.81	−	1679	−	−	15,501,124	Unknown sesquiterpene
28.31	27	1701	n/a	−	15,437,838	1-Isopropyl-4,8-dimethylspiro [4.5]dec-8-en-7-one ^+c^
30.26	−	1793	−	−	21,104,266	Unknown sesquiterpene

**^1^** retention time; **^2^** similarity index; **^3^** Kovats retention index calculated from experimental data; **^4^** Kovats retention index from literature; **^5^** compound exclusively present there (+) or present only in controls (++); **^6^** data not available; **^7^** manual matching; ^a^ compound identified using retention time of authentic standard, matching with MS library and comparison with reported KI; ^b^ compound identified by matching with MS library and comparison with reported KI; ^c^ compound identified by matching with MS library.

**Table 2 molecules-25-05749-t002:** General characteristics of the fungal and oomycete species tested in the experiment.

Treatment	Species	Reference from GenBank
Control	−	−
1	*Armillaria gallica* (Marxm. & Romagn.) 1987	DQ115578 ^1^
2	*Armillaria ostoyae* (Romagn.) Herink 1973	DQ115574
3	*Fusarium avenaceum* (Fr.) Sacc. 1886	MK560761
4	*Fusarium culmorum* (Wm.G. Sm.) Sacc. 1892	KP008988
5	*Fusarium oxysporum* (Schltdl.) 1824	MF162321
6	*Fusarium poae* (Peck) Wollenw. 1913	MF162318
7	*Phytophthora cactorum* (Lebert & Cohn) J. Schröt. 1886	KX242303
8	*Phytophthora cinnamomi* Rands 1922	KF682434
9	*Phytophthora plurivora* T. Jung & T.I. Burgess 2009	JX276032
10	*Phytophthora ramorum* Werres, De Cock & Man in ‘t Veld 2001	JF771575
11	*Rhizoctonia solani* J.G. Kühn 1858	KU901561
12	*Trichoderma asperellum* Samuels, Lieckf. & Nirenberg 1999	MT197117

^1^ number available from www.ncbi.nlm.nih.gov.

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
