# Peer review of "Detection of Fungi and Oomycetes by Volatiles Using E-Nose and SPME-GC/MS Platforms"

_molecules, 2020, doi:10.3390/molecules25235749_

Round 1

Reviewer 1 Report

The paper is really well written. Scientific design is very robust.

I have just some small correction. 

I suggest to uniform the form of fungi name. In lines 34-35 and 573-574-575, family name of fungi is sometime abbreviated and sometime full. Uniforme all. 

Line 190: PDMS can be replaced to DMS.

Line 213. The paragraph number is 2.2.2. and not 3.2.2.

Line 249: Eliminate "A".

Line 295: Replace 107 to 107. 

Lines 545 and 548: VOC's is wrong. Replace with VOCs

Author Response

Reviewer #1

Open Review

English language and style

( ) Extensive editing of English language and style required
( ) Moderate English changes required
(x) English language and style are fine/minor spell check required
( ) I don't feel qualified to judge about the English language and style

Yes

Can be improved

Must be improved

Not applicable

Does the introduction provide sufficient background and include all relevant references?

(x)

( )

( )

( )

Is the research design appropriate?

(x)

( )

( )

( )

Are the methods adequately described?

(x)

( )

( )

( )

Are the results clearly presented?

(x)

( )

( )

( )

Are the conclusions supported by the results?

(x)

( )

( )

( )

Comments and Suggestions for Authors

Answers to the Reviewer #1

We are thankful to the Reviewer for all constructive comments, which helped us to improve the manuscript. Here we present our responses:

R#1: The paper is really well written. Scientific design is very robust.

I have just some small correction. 

I suggest to uniform the form of fungi name. In lines 34-35 and 573-574-575, family name of fungi is sometime abbreviated and sometime full. Uniforme all. 

Answer: Thank you for this remark. We use the full Latin name of the organisms (in Abstract) at first use. Later on, we abbreviate the genus, except in the Table 2, where the full name of the microorganisms tested, where systematics data and reference to the GenBank are given.

R#1: Line 190: PDMS can be replaced to DMS.

Answer: Thank you for this suggestion. It was replaced.

R#1: Line 213. The paragraph number is 2.2.2. and not 3.2.2.

Answer: That is right. It has been corrected to 2.2.2 in line 262.

R#1: Line 249: Eliminate "A".

Answer: Thank you. It has been omitted.

R#1: Line 295: Replace 107 to 107. 

Answer: Thank you for this remark. The order of magnitude was corrected.

Lines 545 and 548: VOC's is wrong. Replace with VOCs

Answer: We agree. It was corrected as suggested.

Reviewer 2 Report

The paper entitled "Detection of fungi and oomycetes by volatiles using e-nose and SPME-GC/MS platforms" is very interesting and it worth to be published in this Journal.  However, there some remarks which are described below:

1) What PDA acronym is? 

2) The abstract must give a preliminary conclusion of the study. Would be good to include the PCA variance.

3) ¿What kind of data pre-processing were applied to the data set by using the E-nose system? for instance: Scaler...

4) A PCA plot Vs Loadings plot should be included in the results in order to know the contributions of original variables to discriminate the fungal species.

5) Information about the data acquisition system, sample collection, and data processing software to obtain results with E-nose should be explained better.

6) Los grupos o categorías de la gráfica PCA (Figura 2) deben identificarse a través de diferentes colores.

7) Please, author's must explain better the differences between GC-MS and E-nose for detecting fungal species. To put a table with the advantages and disadvantages of each one. 

Author Response

Reviewer #2

Answer to the Reviewer #2

We are thankful to the Reviewer for all constructive comments, which helped us to improve the manuscript and present our results in a more clear way. Here we present our responses:

R#2:

The paper entitled "Detection of fungi and oomycetes by volatiles using e-nose and SPE-GC/MS platforms" is very interesting and it worth to be published in this Journal.  However, there some remarks which are described below:

  • What PDA acronym is?

Answer: Thank you for this question. The full name of the “potato dextrose agar” medium was added when first used, i.e. in paragraph 2.1.

  • The abstract must give a preliminary conclusion of the study. Would be good to include the PCA variance.

Answer:  Thank you for this suggestion. We rephrased the end of the Abstract as follows:

Our e-nose system could discriminate between the odors emitted by P. ramorum, F. poae, T. asperellum and R. solani, which accounted for over 88% of the PCA variance. These results make the e-nose device suitable for further sensor design as a potentially suitable tool for forest managers, other plant managers, as well as regulatory agencies such as quarantine services

  • ¿What kind of data pre-processing were applied to the data set by using the E-nose system? for instance: Scaler...

Answer: Thank you for this question. Our idea was to address the problem of diseases observed in Polish forest nurseries Therefore we checked data from advice prepared for forest practitioners based on received samples of soil and plant material. Having the list of the most significant soil borne pathogens (fungi and oomycetes) we then chose “Cultures from 12 different fungi and oomycetes strains (Table 2) kept in the laboratory of Forest Research Institute (IBL).” No more data pre-processing was done for the e-nose system.

  • A PCA plot Vs Loadings plot should be included in the results in order to know the contributions of original variables to discriminate the fungal species.

Answer: Thank you for this suggestion. In the supplementary materials, the correlation plot obtained for the auto-scaled e-nose measurements showing eigenvalues for each principal component was included in Figure S1, as well as in Table S2.

  • Information about the data acquisition system, sample collection, and data processing software to obtain results with E-nose should be explained better.

Answer: Thank you for this comment. The acquisition system was described in fig. 3 and the relevant samples collected from fungi and oomycetes, were described in Table 2. The data processing was described in paragraph 4.2 as follows; “r(j) = R(j) - R0(j) with R(j) representing the averaged measured resistance of the j sensor of the array and R0(j) standing for the averaged baseline value of resistance measured during the blank analysis, both calculated for the j-th sensor of the array based on the last 150 measurements of the corresponding temporal series. Thus, each analyzed sample would yield a final diagnostic feature consisting of a 12-dimensional vector (r1, …, r12) quantifying the reaction of e-nose sensors exposed to the tested material. Overall, twelve such vectors were obtained per treatment, corresponding to the 12 repetitions performed. Calculating the mean of these 12 diagnostic features allowed us to obtain a global averaged signal vector for each treatment, which could be pictured as a diagram. This final 12-dimensional vector was considered as representing the mean e-nose sensor reaction to the treatment, thus providing an assessment of the tested material odor intensity."

  • Los grupos o categorías de la gráfica PCA (Figura 2) deben identificarse a través de diferentes colores.

Answer: We agree with this suggestion. In the PCA chart (Figure 2) in red were marked species of microorganisms with the most pronounced eigenvalues (i.e. > 0.360 on PC1), for P. ramorum (0.369), F. poae (0.409), T. asperellum (0.440), and R. solani (0.496). The other microorganisms were retained in black.

  • Please, author's must explain better the differences between GC-MS and E-nose for detecting fungal species. To put a table with the advantages and disadvantages of each one.

Answer: The aim of our study was not to compare the advantages and disadvantages of E-nose and GC-MS platforms. We believed that those two methods are not substitutable but rather complementary. The application of e-nose served to check the differences in odour among tested organisms while GC-MS analysis served to check the specific compounds for the genus or species level. The idea behind this approach is to replace sensors in the next generation of e-nose devices in the future.

Reviewer 3 Report

In general, the workload of this study is very large, and the contents of results and discussions are rich. But it doesn't give me anything obvious innovative, like instrument design or application scenarios (only identifying the difference of VOCs of different fungi and oomycetes).

The authors do not describe the difference or innovation between this study and previous studies. The specific volatiles of fungi and oomycetes have never been studied before?

What is the relationship between electronic nose and GC-MS in this study? The main differences of VOCs were detected by GC-MS. Electronic noses seem dispensable in this context.

Line 104-105: e-nose could serve as a preliminary screening tool allowing a quick diagnosis that could be achieved even without any symptoms being visible on the plant. That is true, but the early VOCs signals of fungi and oomycetes are usually very weak, and the authors need to simulate the practical level of concentration to see if electronic noses can be used for detection. As shown in Fig. 3a, the concentration of fungi seems significantly higher than that of early infection.

Line 108-109: identification of pathogen-specific VOC profiles could be a key point for improved e-nose sensing efficiency. If the electronic nose used metal oxide gas sensors, in fact there is a lack of a metal oxide gas sensor targeting at a specific VOC. Therefore, even the specific VOCs were found, it will have little effect on the design of the electronic nose using the metal oxide gas sensor.

The authors need to provide the curve of the electronic nose signal during the whole measurement process. Averaging as shown in Fig. 1 might not be the best way to represent the electronic nose signal. In addition, if the fungi and oomycetes are in different concentrations, their electronic nose signals can also be distinguished from other fungi and oomycetes?

Line 277: It should be PCA rather than Principal component analysis

Line 280: I don’t know how to reveal by A SPME-GC/MS?

I don't think it's necessary to separate the results from the discussion, especially with respect to the results of GC-MS, but to merge the results and the discussion with each fungus.

Parts of the description in the paper are too detailed, such as 4.2,  the design idea and detection process of the electronic nose are known, there is no breakthrough, therefore it is not necessary to describe so detailed.

Author Response

Reviewer #3

Answer to the Reviewer #3

We are thankful to the Reviewer for all constructive comments, which helped us to improve the manuscript. Here we present our responses:

R#3: In general, the workload of this study is very large, and the contents of results and discussions are rich. But it doesn't give me anything obvious innovative, like instrument design or application scenarios (only identifying the difference of VOCs of different fungi and oomycetes).

Answer: Thank you for this comment which helped us to improve our manuscript. We have added the following paragraph to the Discussion. The application of an electronic nose in Polish forestry is an innovative idea, as there is no other tool for a quick detection of pathogens such as in forest nurseries. Nowadays the most damaging to plants are soil-borne fungi and oomycetes, the latter ones often the source of emerging diseases (although in stands, forest managers also look every autumn for insects and pests under the canopy of Scots pines). Some pests often cause severe disorders and also need to be controlled. Innovative tool for early detection of pests is crucial for designing an appropriate strategy of their control in the next growing season.

There are more potential applications of such a device such as in ornamental nurseries (similar problems to ones in forestry) and in cities where trees are aging and weakened (often being infected by pathogens), and hence pose a hazard for people and their property (e.g. cars parking along streets). In particular, there is a need for detection of root and butt rot pathogens (like Armillaria spp.) because classical devices like resistographs or sonic tomograph (PICUS) are designed to detect rot in trunks rather than in roots. Early detection of rotten roots by Armillaria may alert managers to remove and replace trees in danger of falling down and to avoid tragedy when people are the victims of such accidents.

Finally, increasing international trade with wood and plant material (potted plants, seeds) allow spread of foreign invasive organisms, including plant pathogens, so e-nose for soil detection in quarantine service could be also very useful.

R#3: The authors do not describe the difference or innovation between this study and previous studies.

Answer: There has been no previous application of the e-nose system for detection of specific volatiles of soil borne fungi and oomycetes in Poland before.

R#3: What is the relationship between electronic nose and GC-MS in this study? The main differences of VOCs were detected by GC-MS. Electronic noses seem dispensable in this context.

Answer: An electronic nose when ready could be a quick and efficient tool for forest managers, especially for production of healthy (pathogen free) seedlings for afforestation and reforestation (replacing traditional not reliable visual selection). The GC-MS in this study helped to find specific compound which could help to replace e-nose sensors for the more efficient ones. It is correct that for the moment the main differences of VOCs were detected by GC-MS. In this context an electronic nose will be a dispensable tool in the future as GC-MS methods are applicable for laboratory only and are time and work demanding. Foresters need a portable tool for quick detection of potential forest pests both in nurseries, plantations and adult stands, and GC-MS does not have a portable field use platform.

R#3: Line 104-105: e-nose could serve as a preliminary screening tool allowing a quick diagnosis that could be achieved even without any symptoms being visible on the plant. That is true, but the early VOCs signals of fungi and oomycetes are usually very weak, and the authors need to simulate the practical level of concentration to see if electronic noses can be used for detection. As shown in Fig. 3a, the concentration of fungi seems significantly higher than that of early infection.

Answer: That is correct, that is why our next attempt will concentrate on soil samples put (similarly like fungi and oomycetes) into glass jars. We believe that VOCs concentration above soil in jars will be detectable by new generation of e-noses within of 20-30 minutes. Appropriate statement was added in the paragraph 3.1 as follows: “Unfortunately, the early VOCs signals of fungi and oomycetes are usually very weak, and in the future we need to simulate the practical level of concentration to see if electronic noses can be used for detection. As shown in Fig. 3a, the concentration of fungi seems significantly higher than that of early infection. Therefore, in next experiments we will concentrate on soil samples, which we will put immediately in nurseries into glass jars (similarly like fungi and oomycetes). We believe that in jars VOCs concentration above the soil will be detectable by new generation of e-noses within of 20-30 minutes, which is enough from practical view.”

R#3: Line 108-109: identification of pathogen-specific VOC profiles could be a key point for improved e-nose sensing efficiency. If the electronic nose used metal oxide gas sensors, in fact there is a lack of a metal oxide gas sensor targeting at a specific VOC. Therefore, even the specific VOCs were found, it will have little effect on the design of the electronic nose using the metal oxide gas sensor.

Answer: The future e-nose will be using pair of sensors as much as possible specific to the investigated group of organisms. From the figure 1 we can see that sensor No 8 and 9 always give the lowest response so we conclude that they should be replaced by others. GC-MS technique can help us to choose specific compounds for which sensors could be designed. However in the new generation of e-noses we would like to apply artificial intelligence (neural networks) but in order to learn them we will need to feed them with numerous data base. In such a case will be not important the identification of particular compound(s) but the specific pattern of their occurrence (like in the image).

R#3: The authors need to provide the curve of the electronic nose signal during the whole measurement process. Averaging as shown in Fig. 1 might not be the best way to represent the electronic nose signal. In addition, if the fungi and oomycetes are in different concentrations, their electronic nose signals can also be distinguished from other fungi and oomycetes?

Answer: At this stage we wanted to know first of all if there is a difference in odour (VOC’s) among chosen organisms. We have learned that some of them smell less than other even within Phytophthora group P. ramorum gave much higher signals than other oomycetes. It is promising because this species is quarantine one and important not let her enter into new area. We thought that in Europe it can be dangerous for oaks (like in US causing sudden oaks death- SOD phenomenon) but amazingly it attacked larch (first Japanese species and now European). According to Brasier (paper in Nature) thousands of hectares and millions of trees are affected and dying). Above information let us to check what are the chemical compounds produced by investigated organisms (including P. ramorum) using GC-MS method. This learning will serve to modify and build a new generation of e-noses better discriminating signals (measurements of resistance) not only of fungal and oomycetes pure cultures but also plants infected by those pathogenic organisms.

R#3: Line 277: It should be PCA rather than Principal component analysis

Answer: The full name of this analysis was shortened to PCA in the sentence “Principal component analysis of sensorial measurements under laboratory conditions…” and changed to “PCA of sensorial measurements under laboratory conditions…”

R#3: Line 280: I don’t know how to reveal by A SPME-GC/MS?

I don't think it's necessary to separate the results from the discussion, especially with respect to the results of GC-MS, but to merge the results and the discussion with each fungus.

Parts of the description in the paper are too detailed, such as 4.2, the design idea and detection process of the electronic nose are known, there is no breakthrough, therefore it is not necessary to describe so detailed.

Answer: We would like to keep the layout of the paper as we believe it easier to follow by the reader if the paragraphs are shorter and specific to the given topic (e.g. fungus). Concerning detailed description of the e-nose in the paragraph 4.2 we also would like to keep it because our foresters asked for it, as it is still a new a technique in Polish forestry. Moreover, new e-noses, which are going to build will be designed differently in order to improve their features after taking into account the present lessons. We will probably give up from 2 pipes and an approach comparing ambient air with tested material but we will combine it within one sensor chamber. In other words, there will be no longer 2 chambers with sensors in the new e-nose. But there will be a large and cylinder with reference clean air (ambient air) and an air / fragrance switch.

Reviewer 4 Report

The manuscript describes the application of both electronic nose and GC/MS for the characterization of the volatiles emitted by fungi. The topic could be interesting, but there are some flaws in the way the measurements have been carried out. The sampling procedure is very critical for the measurements with electronic noses and the unusual presence of volatiles even in the presence of empty bottle or with PDA medium can be explicated looking at Figure 3. Plastic tubes and o-ring of course induce the presence of artifacts that should be avoided. Furthermore the air flux of 1 L/min seems to be quite high, considering the small volume of the container: if the measurement last 5 minutes, the headspace of fungi is completely consumed and this results in a flux measurement. A figure with sensor responses will be helpful to evaluate the influence in the results obtained.

Furthermore the PC1 (Figure 2) seems to be strongly correlated with the sensor responses and consequently with volatile concentrations, which, considering the problems with the sampling procedures, arises doubts on the real discrimination ability of the sensor array.

The identification of the volatiles by GC/MS has been carried out using the NIST library, but support to the identification by internal standard method is necessary.

Furthermore the link between electronic nose and GC/MS characterization is not clear and they seem two separate works: the discrimination operated by electronic nose depends on the different composition of the fungi headspace or to the amount of volatiles?

Author Response

Reviewer #4

Open Review

English language and style

( ) Extensive editing of English language and style required
( ) Moderate English changes required
(x) English language and style are fine/minor spell check required
( ) I don't feel qualified to judge about the English language and style

Yes

Can be improved

Must be improved

Not applicable

Does the introduction provide sufficient background and include all relevant references?

(x)

( )

( )

( )

Is the research design appropriate?

( )

( )

(x)

( )

Are the methods adequately described?

( )

( )

(x)

( )

Are the results clearly presented?

( )

( )

(x)

( )

Are the conclusions supported by the results?

( )

( )

(x)

( )

Comments and Suggestions for Authors

Answer to the Reviewer #4

We are thankful to the Reviewer for all constructive comments, which helped us to improve the manuscript. Here we present our responses:

R#4: The manuscript describes the application of both electronic nose and GC/MS for the characterization of the volatiles emitted by fungi. The topic could be interesting, but there are some flaws in the way the measurements have been carried out. The sampling procedure is very critical for the measurements with electronic noses and the unusual presence of volatiles even in the presence of empty bottle or with PDA medium can be explicated looking at Figure 3. Plastic tubes and o-ring of course induce the presence of artifacts that should be avoided. Furthermore the air flux of 1 L/min seems to be quite high, considering the small volume of the container: if the measurement last 5 minutes, the headspace of fungi is completely consumed and this results in a flux measurement. A figure with sensor responses will be helpful to evaluate the influence in the results obtained.

Answer: Thank you for raising this point. We tried to avoid as much as possible all artefacts, that is why we changed plastic Petri dishes to glass jars (for pure cultures), and used chemically inert substances when possible (tubes and o-rings). In addition we always compared treatment results to the control and tried to maintain the same conditions of the experiment. However, we agree with the comment, and in the future we will change the approach of comparing ambient air with the tested one. In such a way the tubes will be eliminated and we will build one sensor chamber only with control temperature and humidity. We will increase the time of measurements too (e.g. 30 min) with the same time for breaks between measurements. We also will consider using clean chambers with measurements between treatment or sampling. In this first experiment we just wanted to know whether it is possible to deal with soil borne pathogens. Are they giving different signals among them and control? We learned that P. ramorum produced a higher signal from among all investigated oomycetes (Fig. 1), and with GC-MS method we found the compounds occurring only in its cultures, which are probably responsible for it. Similarly, we analysed other groups e.g. Fusaria and found that F. poae showed the strongest signal. These are the first and encouraging investigations with forest soil borne pathogens, which will be further developed in nurseries (practical context). Foresters cannot allow the diseased seedlings to leave nursery and be planted in forest plantations. Visual inspection often fails so they are happy if such a new tool will help them to detect pathogens at an early step of production of reproductive material.

R#4: Furthermore the PC1 (Figure 2) seems to be strongly correlated with the sensor responses and consequently with volatile concentrations, which, considering the problems with the sampling procedures, arises doubts on the real discrimination ability of the sensor array.

Answer: Thank you for raising this point. In fact the 1st component (PC1) explains almost all (88.2%) of variability between sensors. That is why we estimated four microorganisms (F. poae, R. solani, T. asperellum and P. ramorum) as the most distinguished ones, i.e. with the most pronounced eigenvalues.

R#4: The identification of the volatiles by GC/MS has been carried out using the NIST library, but support to the identification by internal standard method is necessary.

Answer: In our study the molecular identification was caried out using threefold approach: matching with GC/ES MS libraries, both commercial (Willey) and in-house built, comparison of Kovats indices (measured + reported elsewhere) and authenticated standards. When it comes to the last case, this method was applied if only standards were available, and thus was confined to common organics, such as hexanal, heptanal or benzaldehyde. However, a few unknowns from fungal emissions, e.g., 1,3,4,5,6,7-hexahydro-2,5,5-trimethyl-2H-2,4a-ethanonaphthalene, where tentatively elucidated using two first approaches since they lacked  authenticated standards. We agree with the reviewer that identification with standard (authentic not internal) is relevant for the thorough identification. However, the approach shows severe limitations owing to availability of synthetic standards.

R#4: Furthermore the link between electronic nose and GC/MS characterization is not clear and they seem two separate works: the discrimination operated by electronic nose depends on the different composition of the fungi headspace or to the amount of volatiles?

Answer: We concur with the reviewer that it is relevant to set up a link between electronic nose and GC/MS characterization and some additional information was added in the discussion. In our research we decided to apply a two-step approach: first to check whether the developed detection system for fungi and fungi-like organisms (called e-nose) is possible to apply in order to distinguish them on the genus or species level, and second to identify the released molecules through solid phase microextraction–gas chromatography/mass spectrometry (SPME–GC/MS) analysis. When it comes to the e-nose, the operation was executed using a block approach, when different chemicals belonging to the same chemical class, e.g. hydrocarbons, aldehydes, etc. were researched. However, a number of these chemicals were identified by GC/MS-induced analysis. In other words, we learned from the e-nose approach that some organisms have more odors than others, and GC-MS analysis was used to showed us why. We found chemical compounds being specific only for particular organisms, which we believe could serve for better designing sensors in the next generation of e-noses.

Round 2

Reviewer 3 Report

Thank you for your careful and detailed reply.

I think it's very difficult to use electronic noses in forestry to detect pathogens currently. Electronic nose testing involves placing the sample in an airtight container for a period of time. The author needs to make this point in the paper.

The innovations explained by the author are actually meaningful, which is very useful, but for the researchers of VOC detection, there is no obviously innovation in terms of technology and methods.

Again, not in Poland is not enough to indicate the novelty of this paper. Molecules is a global journal.

As for the explanation of electronic nose and GC-MS, I know the advantages and disadvantages of both. However, in this study, the research between the two technologies has not been connected well. The author should have described this more. Also, it is a correct research idea to develop special electronic nose apparatus after finding characteristic compounds by GC-MS, but it is not easy to develop specific electronic nose apparatus.

It is incorrect to use the simulated concentration to show that the actual concentration can be detected. The results of this paper can only say that electronic nose can be used to detect simulated concentration. In addition, 20-30 minutes of testing is a long time, not suitable for practical application. The simulated concentration should be as close as possible to the actual concentration.

If the detection is carried out by a specific pattern, rather than by characteristic compounds, then the meaning of conducting GC-MS research is not significant.

The author did not answer directly about the curve of the electronic nose signal. The authors need to provide the curve of the electronic nose signal during the whole measurement process.

For the sentence "A SPME/GC/MS revealed that sensors 1 and 2 were receptive to ethanol", my question is that I don't know how to reveal by SPME/GC/MS that sensors 1 and 2 were associated with ethanol. The author did not answer this question.

To my last question, the Molecules is a global journal, and its readers are not only from Poland. I think the current layout is not easy to read. Meanwhile, much of what is known, such as the current second data on the design idea and detection process of the electronic nose, need not be too detailed. This can be done by citing references. If the readers need further information, they can consult these references.

Author Response

Answer to the Reviewer remarks and to the Academic Editor Notes

We are thankful to the Reviewer for all constructive comments, which helped us to basically revise and improve the manuscript. Here we present our responses to all comments from our academic editor, guest editor and the reviewer:
"After reading the manuscript and the comments of the reviewers I agree to reviewer 3 that the manuscript is not suitable for publication. I agree to the comments of reviewer 3, particularly in report 2.

Response: We carefully examined the questionable data and improved them. We better explained the design of the study and provided missing information.

“Moreover, I am wondering that no replicates were done (at least that was not described in the manuscript). I would, for instance, expect error bars in Fig. 1 calculated from independent experiments and several points in the PCA diagram shown in Fig. 2, which should – hopefully – cluster.

Response: Thank you for this comment we apologize that in the description of methods we skipped this issue unintentionally. We hope that now, after major revision of the manuscript and addressing all the comments of the reviewer 3, it is clear and the manuscript is suitable for publication. We stressed replicates, which were performed (now they are described in lines 607-611 and 626-627). We also replaced Fig. 1, now with error bars.

The paragraph (4.4.) about statistical analyses was completely rewritten and new calculations have been done. The variability among signals obtained from twelve sensors of e-nose device was computed by principal component analysis (PCA) in “R” software (R Core Team, 2015). PCA analyses, as well as the biplot were created with fviz_pca_biplot functions from the “FactoMineR” 1.41 package (Husson et al., 2015). The variables with the strongest impact on the distribution of the microorganisms along the principal components were identified on the basis of Pearson correlation coefficients (Fig. 2 and Table S2).  

To group microorganisms according to their signal similarity among twelve sensors, a hierarchical clustering using Euclidean distance (root sum-of-squares of differences) as similarity measure and Ward (1963) clustering method with the criterion proposed by Murtagh and Legendre (2014) was applied. Four different clustering methods, single and complete linkage, the unweighted pair group method with arithmetic mean (UPGMA) and Ward’s method were tested according to their clustering structure of the dataset (Kaufman and Rousseeuw, 2018). Finally, Ward’s method was chosen, since it expressed the highest value of the agglomerative coefficient. Hierarchical clustering was performed using the Cluster 2.0.7-1 package (Rousseeuw et al. 2018). Final grouping of microorganisms according to their locations on the PCA biplot were determined visually (Fig. 2).

This served to improve the PCA diagram shown in new Fig. 2. Based on correlation analyses between eigenvalues, we found that the variance between signals mostly depends on the first principal component (PC1) and only slightly on the second principal component (PC2) (93.87% and 03.26%, respectively; Figure 2, Table S2). This enabled us to uncover from the PCA plot that the odors generated by P. ramorum, F. poae, T. asperellum and R. solani were readably distinguishable based on the location of the microorganisms on the biplot and grouping pattern confirmed by hierarchical clustering analysis (Figure S1, Table S2).

“In addition, the design of the study is questionable. If developing an electronic nose for detection of fungal infections in forestry it is not useful investigating VOC produced by fungi grown on synthetic media. Instead infected plants, ideally different species, should be tested. In any case, even if this is only a preliminary study and use of synthetic media is considered as acceptable at least different synthetic media and cultures of different age should be measured to see how robust identification is.

Response: In order to develop an electronic nose for detection of fungal infections in forestry we have started by investigating VOC produced by fungi growing on synthetic media to understand whether they produce specific compounds, which could be used as sensors during developing of our device. In the next stage, plants will be inoculated with those species, which are top target by foresters and quarantine officers for early detection in nurseries. As it is only a preliminary study we tested different synthetic media for cultivation and control and chose one, which is broadly used in forest pathology laboratories. We also eliminated the use of plastic Petri dishes during our preliminary tests, and replaced them with glass plates which should be chemically neutral. Finally, we manually checked odour of cultures of different age (from 1 week to 4 weeks) and sometimes noted changes in odor which were verified with the e-nose to make such measurements robust and amenable to statistical analysis.

However, even working on pure cultures, e-nose, as a tool, could be applicable in forest nurseries for a preliminary selection of infected plants. We have been using baiting techniques to obtain pure cultures of pathogenic fungi and oomycetes from tissues such as infected oak or beech leaves. Usually, cultures are further transported to the lab for DNA extraction and molecular analysis. To shorten this procedure, we can use the e-nose to distinguish oomycetes from fungi, and design appropriate methods of control. Most fungicides against plant diseases do not work for oomycetes since they are more often targeted to ascomycetous pathogens. Pesticides used in forest nurseries can mask diseases, which can then emerge after transplanting into forests or into riparian ecosystems, where it is then difficult and costly to manage their development in situ. Therefore, the nursery is a critcal location to detect and separate diseased plant material (of select pathogens) from health plants, and the e-nose could be developed for feasible assessment in the future.

“The presented data do not allow any conclusion about that. According to the methods VOCs were only identified in GC-MS by the mass spectrum while other parameters, particularly the retention time were not taken into account. This is a very questionable practice and can lead to wrong identification. I would expect the authors confirm at least some of the “identified” compounds by running authentic standards.

Response: The presented data were enriched by retention time and now do allow us to draw the preliminary conclusion that investigated organisms differed, and the signals registered by 12 sensors showed that some of them are more odoriferous than others meaning that they should be distinguishable based on this feature and hopefully in the future, we will be able to identify them based on the differences in VOC synthesis.

To address reviewer 3 comments, we added new paragraphs in sections 2.2 and 4.3. 

The molecular identification was carried out using a threefold approach: matching with GC/EI MS libraries, both commercial (NIST + Willey) and in-house built; comparison of Kovats indices (measured + reported elsewhere); and comparison with authentic standards. The last approach was applied only when standards were available, and thus was confined to common organics, such as hexanal, heptanal or benzaldehyde. However, a few unknowns from fungal emissions, e.g., 1,3,4,5,6,7-hexahydro-2,5,5-trimethyl-2H-2,4a-ethanonaphthalene, were tentatively elucidated using first two  approaches since the lack of their authentic standards. The identification with standards (authenticated not internal) was the gold standard for thorough identification, but the approach was limited by access or availability of synthetic standards.

To identify components, both mass spectral data and the calculated retention indices were used. Mass spectrometric identification not confirmed by the retention index was considered as putative.

In the paragraph 4.3: we added information about NIST Mass Spectral Database and Willey libraries for the analyte identification. In addition, for some unknowns the identification was supported by in-house constructed library comprising the EI mass spectra for available standards.

“For peaks not confirmed by standards “identification” based solely on MS spectra must be described as tentative."

Response: According to the applied GC-MS method VOCs were first identified by the retention time and later by the mass spectrum. This is widely acceptable practice leading to identification with high confidence. However, we have not confirmed all found compounds by running authenticated standards, yet. For this reason peaks not confirmed by standards but based solely on MS spectra are considered tentative.

That is why we also stress in the abstract of our manuscript preliminary fungi and oomycete detection by volatiles using e-nose and SPME-GC/MS platforms.

Reviewer 4 Report

The manuscript has been improved and almost all my concerns have been satisfied by the authors replies.

I understand that this is an explorative work, so most of the experimental details could be improved in the near future.

In my opinion the manuscript can be now accepted for publication.

Author Response

Dear Editor in Chief,

We are very thankful for all constructive comments, which helped us to improve the final version of the manuscript. We followed the comments and clearly identified compounds based on either (1) an authentic standard (under identical conditions), (2) MS spectra and RI taken from the literature or (3) tentatively identified solely by the MS spectrum. We used superscript symbols (1, 2 or 3) to tag the compound and explained the meaning in the legend below the table.

We also updated the results section by mentioning that, in addition to the MS spectra, also RI and authentic standards were used for identification of the compounds.

We hope that the results will be interesting for the readers of the Molecules Journal.

Thank you very much for your excellent efforts.

Yours sincerely

Justyna Nowakowska

Professor

University of Cardinal Stefan Wyszynski

e-mail: J.Nowakowska@uksw.edu.pl